# NECO: NEural Collapse Based Out-of-distribution detection

**Mouïn Ben Ammar**◇,†,*, **Nacim Belkhir**†, **Sebastian Popescu**◇, **Antoine Manzanera**◇, **Gianni Franchi**◇ *

U2IS Lab ENSTA Paris◇, Palaiseau, FRANCE

SafranTech†, Chateaufort 78117, FRANCE

`{first.last}@enstaparis.com`◇, `safrangroup.com`†

## Abstract

Detecting out-of-distribution (OOD) data is a critical challenge in machine learning due to model overconfidence, often without awareness of their epistemological limits. We hypothesize that "neural collapse", a phenomenon affecting in-distribution data for models trained beyond loss convergence, also influences OOD data. To benefit from this interplay, we introduce NECO, a novel post-hoc method for OOD detection, which leverages the geometric properties of "neural collapse" and of principal component spaces to identify OOD data. Our extensive experiments demonstrate that NECO achieves state-of-the-art results on both small and large-scale OOD detection tasks while exhibiting strong generalization capabilities across different network architectures. Furthermore, we provide a theoretical explanation for the effectiveness of our method in OOD detection. Code is available at `https://gitlab.com/drti/neco`.

## 1 Introduction

In recent years, deep learning models have achieved remarkable success across various domains (OpenAI, 2023; Ramesh et al., 2021; Jumper et al., 2021). However, a critical vulnerability often plagues these models: they tend to exhibit unwarranted confidence in their predictions, even when confronted with inputs that deviate from the training data distribution. This issue gives rise to the challenge of Out-of-Distribution (OOD) detection for Deep Neural Networks (DNNs). OOD detection holds significant safety implications. For instance, in medical imaging a DNN may fail to make an accurate diagnosis when presented with data that falls outside its training distribution (*e.g.*, using a different scanner). A reliable DNN classifier should not only correctly classify known In-Distribution (ID) samples but also flag any OOD input as "unknown". OOD detection plays a crucial role in ensuring the safety of applications such as medical analysis (Schlegl et al., 2017), industrial inspection (Paul Bergmann & Stege, 2019), and autonomous driving (Kitt et al., 2010).

There are various approaches to distinguish ID and OOD data that fall into three main categories: confidence-based (Liu et al., 2020; Hendrycks & Gimpel, 2017; Hendrycks et al., 2022; Huang et al., 2021; Liang et al., 2018), features/logits-based (Sun & Li, 2022; Sun et al., 2021; Wang et al., 2022; Djurisic et al., 2022) and distance/density based (Ming et al., 2023; Lee et al., 2018b; Sun et al., 2022). OOD detection can be approached in a supervised manner, primarily employing outlier exposure methods (Hendrycks et al., 2018), by training a model on OOD datasets. However, in this work, we focus on post-hoc (unsupervised) OOD detection methods. These methods do not alter the network training procedure, hence avoid harming performance and increasing the training cost. Consequently, they can be seamlessly integrated into production models. Typically, such methods leverage a trained network to transform its latent representation into a scalar value that represents the confidence score in the output prediction. The underlying presumption is that ID samples should yield high-confidence scores, while the confidence should notably drop for OOD samples. Post-hoc approaches make use of a model-learned representation, such as the model's logits or deep features which are typically employed for prediction, to compute the OOD score.

A series of recent studies (Ishida et al., 2020; Papyan V, 2020) have shed light on the prevalent practice of training DNNs well beyond the point of achieving zero error, aiming for zero loss. In the

---

*corresponding author

Terminal Phase of Training (TPT), occurring after zero training set error is reached, a "Neural Collapse" (NC) phenomenon emerges, particularly in the penultimate layer and in the linear classifier of DNNs (Papyan V, 2020), and it is characterized by four main properties:

1. **Variability Collapse (NC1):** during the TPT, the within-class variation in activations becomes negligible as each activation collapses toward its respective class mean.

2. **Convergence to Simplex ETF (NC2):** the class-mean vectors converge to having equal lengths, as well as having equal-sized angles between any pair of class means. This configuration corresponds to a well-studied mathematical concept known as Simplex Equiangular Tight Frame (ETF).

3. **Convergence to Self-Duality (NC3):** in the limit of an ideal classifier, the class means and linear classifiers of a neural network converge to each other up to rescaling, implying that the decision regions become geometrically similar and the class means to lie at the centers of their respective regions.

4. **Simplification to Nearest Class-Center (NC4):** the network classifier progressively tends to select the class with the nearest class mean for a given activation, typically based on standard Euclidean distance.

These NC properties provide valuable insights into how DNNs behave during the TPT. Recently it was demonstrated that collapsed models exhibit improved OOD detection performance (Haas et al., 2023). Additionally, they found that applying L2 normalization can accelerate the model's collapse process. However, to the best of our knowledge, no one has yet evidenced the following interplay between NC of ID and OOD data:

5. **ID/OOD Orthogonality (NC5):** As the training procedure advances, OOD and ID data tend to become increasingly more orthogonal to each other. In other words, the clusters of OOD data become more perpendicular to the configuration adopted by ID data (*i.e.*, the Simplex ETF).

Building upon the insights gained from the aforementioned properties of NC, as well as the novel observation of ID/OOD orthogonality (NC5), we introduce a new OOD detection metric called "NECO", which stands for NEural Collapse-based Out-of-distribution detection. NECO involves calculating the relative norm of a sample within the subspace occupied by the Simplex ETF structure. This subspace preserves information from ID data exclusively and is normalized by the norm of the full feature vector.

We summarize our contributions as follows:

- We introduce and empirically validate a novel property of NC in the presence of OOD data.

- We proposed a novel OOD detection method **NECO**, a straightforward yet highly efficient post-hoc method that leverages the concept of NC. Furthermore, we offer a comprehensive theoretical analysis that sheds light on the underlying mechanisms of **NECO**.

- NECO demonstrates superior performance compared to state-of-the-art methods across different benchmarks and architectures, such as ResNet-18 on CIFAR10/CIFAR100, and vision transformer networks on ImageNet-1K.

## 2 RELATED WORK

**Neural Collapse**   is a set of intriguing properties that are exhibited by DNNs when they enter the TPT. NC in its essence, represents the state at which the within-class variability of the penultimate layer outputs collapse to a very small value. Simultaneously, the class means collapse to the vertices of a Simplex ETF. This empirically emergent structure simplifies the behavior of the DNN classifier. Intuitively, these properties depict the tendency of the network to maximize the distance between the class cluster mean vectors while minimizing the discrepancy within a given cluster. Papyan V (2020) have shown that the collapse property of the model induces generalization power and adversarial robustness, which persists across a range of canonical classification problems, on different neural network architectures (*e.g.*, VGG (Simonyan & Zisserman, 2015), ResNet (He et al., 2016),

and DenseNet (Huang et al., 2017) and on a variety of standard datasets (*e.g.*, MNIST (Deng, 2012), CIFAR-10 and CIFAR-100 (Krizhevsky, 2012)), and ImageNet (Russakovsky et al., 2015)). NC behavior has been empirically observed when using either the cross entropy (Papyan V, 2020) or the mean squared error (MSE) loss (Han et al., 2022). Many recent works attempt to theoretically analyze the NC behavior (Yang et al., 2022; Kothapalli, 2023; Ergen & Pilanci, 2021; Zhu et al., 2021; Tirer & Bruna, 2022), usually using a mathematical framework based on variants of the unconstrained features model, proposed by Mixon et al. (2020).

Haas et al. (2023) states that collapsed models exhibit higher performance in the OOD detection task. However, to our knowledge, no one has attempted to directly leverage the emergent properties of NC to the task of OOD detection.

**OOD Detection**   has attracted a growing research interest in recent years. It can be divided into two pathways: supervised and unsupervised OOD detection. Due to the post-hoc nature of our method, we will focus on the latter. Post-hoc approaches can be divided into three main categories. Firstly, confidence-based methods. These methods utilize the network final representation to derive a confidence measure as an OOD scoring metric (DeVries & Taylor, 2018; Huang & Li, 2021b; Hendrycks & Gimpel, 2017; Liu et al., 2020; Hendrycks et al., 2022; Liang et al., 2018; Huang et al., 2021). Softmax score (Hendrycks & Gimpel, 2017) is the common baseline for post-hoc methods as they use the model softmax prediction as the OOD score. Energy (Liu et al., 2020) elaborates on that principle by computing the energy (*i.e.*, the logsumexp on the logits), with demonstrated advantages over the softmax confidence score both empirically and theoretically. ODIN (Liang et al., 2018) enhances the softmax score by perturbing the inputs and rescaling the logits. Secondly, distance/density based (Abati et al., 2019; Lee et al., 2018a; Sabokrou et al., 2018; Zong et al., 2018; Lee et al., 2018b; Ming et al., 2023; Ren et al., 2021; Sun et al., 2022; Techapanurak et al., 2019; Zaeemzadeh et al., 2021; van Amersfoort et al., 2020). These approaches identify OOD samples by leveraging the estimated density on the ID training samples. Mahalanobis (Lee et al., 2018b) utilizes a mixture of class conditional Gaussians on the distribution of the features. (Sun et al., 2022) uses a non-parametric nearest-neighbor distance as the OOD score. Finally, the feature/logit based methods utilize a combination of the information within the model's logits and features to derive the OOD score. (Wang et al., 2022) utilizes this combination to create a virtual logit to measure the OOD nature of the sample. ASH (Djurisic et al., 2022) utilizes feature pruning/filling while relying on sample statistics before passing the feature vector to the DNN classifier. Our method lies within the latter subcategory, bearing different degrees of similarity with some of the methods. In the same subcategory, methods like NuSA (Cook et al., 2020) or ViM (Wang et al., 2022) leverage the principal/Null space to compute their OOD metric. More details are presented at 4.1.

## 3   PRELIMINARIES

### 3.1   BACKGROUND AND HYPOTHESES

In this section, we will establish the notation used throughout this paper. We introduce the following symbols and conventions:

- We represent the training and testing sets as $D_l = (\boldsymbol{x}_i, y_i)_{i=1}^{n_l}$ and $D_\tau = (\boldsymbol{x}_i, y_i)_{i=1}^{n_\tau}$, respectively. Here, $\boldsymbol{x}_i$ represents an image, $y_i \in [\![0, C]\!]$ denotes its associated class identifier, and $C$ stands for the total number of classes. It is assumed that the data in both sets are independently and identically distributed (i.i.d.) according to their respective unknown joint distributions, denoted as $\mathcal{P}_l$ and $\mathcal{P}_\tau$.

- In the context of anomaly detection, assumeion that $\mathcal{P}_l$ and $\mathcal{P}_\tau$ exhibit a high degree of similarity. However, we also introduce another test dataset denoted as $D_{\text{OOD}} = (\boldsymbol{x}_i^{\text{OOD}}, y_i^{\text{OOD}})_{i=1}^{n_{\text{OOD}}}$, where the data is considered to be i.i.d. according to own unknown joint distribution, referred to as $\mathcal{P}_{\text{OOD}}$, which is distinct from both $\mathcal{P}_l$ and $\mathcal{P}_\tau$.

- The DNN is characterized by a vector containing its trainable weights, denoted as $\boldsymbol{\omega}$. We use the symbol $f$ to represent the architecture of the DNN associated with these weights, and $f_{\boldsymbol{\omega}}(\boldsymbol{x}_i)$ denotes the output of the DNN when applied to the input image $\boldsymbol{x}_i$.

- To simplify the discussion, we assume that the DNN can be divided into two parts: a feature extraction component denoted as $h_{\boldsymbol{\omega}}(\cdot)$ and a final layer, which acts as a classifier and is

denoted as $g_{\boldsymbol{\omega}}(\cdot)$. Consequently, for any input image $\boldsymbol{x}_i$, we can express the DNN's output as $f_{\boldsymbol{\omega}}(\boldsymbol{x}_i) = (g_{\boldsymbol{\omega}} \circ h_{\boldsymbol{\omega}})(\boldsymbol{x}_i)$.

- In the context of image classification, we consider the output of $h_{\boldsymbol{\omega}}(\cdot)$ to be a vector, which we denote as $\boldsymbol{h}_i = h_{\boldsymbol{\omega}}(\boldsymbol{x}_i) \in \mathbb{R}^D$ for image $\boldsymbol{x}_i$ with $D$ the dimension of the feature space.

- We define the matrix $\mathbf{H} \in M_{n_l, D}(\mathbb{R})$ as containing all the $h_{\boldsymbol{\omega}}(\boldsymbol{x}_i)$ values where $\boldsymbol{x}_i$ belongs to the training set $D_l$. Specifically, $\mathbf{H} = [h_{\boldsymbol{\omega}}(\boldsymbol{x}_1) \quad \dots \quad h_{\boldsymbol{\omega}}(\boldsymbol{x}_{n_l})]$ represents the feature space within the ID data.

- We introduce $D_l^c$ as a dataset consisting of data points belonging to class $c$, and $\mathbf{H}^c$ represents the feature space for class $c \in [\![0, C]\!]$.

- For a given combination of dataset and DNN, we define the empirical global mean $\mu_G = 1/\text{card}(D_l) \sum_{\boldsymbol{x}_i \in D_l} h_{\boldsymbol{\omega}}(\boldsymbol{x}_i) \in \mathbb{R}^D$ and the empirical class means $\mu_c = 1/\text{card}(D_l^c) \sum_{\boldsymbol{x}_i \in D_l^c} h_{\boldsymbol{\omega}}(\boldsymbol{x}_i) \in \mathbb{R}^D$, where $\text{card}(\cdot)$ represents the number of elements in a dataset.

- In the context of a specific dataset and DNN configuration, we define the empirical covariance matrix of $\mathbf{H}$ to refer to $\Sigma_T \in M_{D \times D}(\mathbb{R})$. This matrix encapsulates the total covariance and can be further decomposed into two components: the between-class covariance, denoted as $\Sigma_B$, and the within-class covariance, denoted as $\Sigma_W$. This decomposition is expressed as $\Sigma_T = \Sigma_B + \Sigma_W$.

- Similarly for a given combination of an OOD dataset and a DNN, we define the OOD empirical global mean $\mu_G^{\text{OOD}}$ and the OOD empirical class means $\mu_c^{\text{OOD}}$, and the OOD feature matrix $\mathbf{H}^{\text{OOD}} \in M_{n_{\text{OOD}}, D}(\mathbb{R})$.

In the context of unsupervised OOD detection with a post-hoc method, we train the function $f_{\boldsymbol{\omega}}(\cdot)$ using the dataset $D_l$. Following the training process, we evaluate the performance of $f_{\boldsymbol{\omega}}$ on a combined dataset consisting of both $D_{\text{OOD}}$ and $D_\tau$. Our objective is to obtain a confidence score that enables us to determine whether a new test data point originates from $D_{\text{OOD}}$ or $D_\tau$.

## 3.2 NEURAL COLLAPSE

Throughout the training process of $f_{\boldsymbol{\omega}}$, it has been demonstrated that the latent space, represented by the output of $h_{\boldsymbol{\omega}}$, exhibits four distinct properties related to NC. In this subsection we delve deeper into the first two, with the remaining two being detailed in D. **The first Neural Collapse (NC1)** property is related to the Variability Collapse of the DNN. As training progresses, the within-class variation of the activations diminishes to the point where these activations converge towards their respective class means, effectively making $\Sigma_W$ approach zero. To evaluate this property during training, (Papyan V, 2020) introduced the following operator:

$$\text{NC1} = \text{Tr}\left[\frac{\Sigma_W \Sigma_B^\dagger}{C}\right] \qquad (1)$$

Here, $[.]^\dagger$ signifies the Moore-Penrose pseudoinverse. While the authors of Papyan V (2020) state that the convergence of $\Sigma_W$ towards zero is the key criterion for satisfying NC1, they also point out that $\text{Tr}\left[\Sigma_W \Sigma_B^\dagger\right]$ is commonly employed in multivariate statistics for predicting misclassification. This metric measures the inverse signal-to-noise ratio in classification problems. This formula is adopted since it scales the intra-class covariance matrix $\Sigma_W$ (representing noise) by the pseudoinverse of the inter-class covariance matrix $\Sigma_B$ (representing signal). This scaling ensures that NC1 is expressed in a consistent reference frame across all training epochs. When NC1 approaches zero, it indicates that the activations are collapsing towards their corresponding class means.
**The second Neural Collapse (NC2)** property is associated with the phenomenon where the empirical class means tend to have equal norms and to spread in such a way as to equalize angles between any pair of class means as training progresses. Moreover, as training progresses, these class means tend to maximize their pairwise distances, resulting in a configuration akin to a Simplex ETF. This property manifests during training through the following conditions:

Here, $\| \cdot \|_2$ represents the L2 norm of a vector, $| \cdot |$ denotes the absolute value, $\langle \cdot, \cdot \rangle$ is the inner product, and $\delta_{..}$ is the Kronecker delta symbol. The convergence to the Simplex ETF is assessed

through two metrics that each verify the following properties: the "equinormality" of class/classifier means, along with their "Maximum equiangularity". Equinormality of class means is measured using its variation, as follows:

$$\text{EN}_{\text{class-means}} = \frac{\text{std}_c\left\{\|\mu_c - \mu_G\|_2\right\}}{\text{avg}_c\left\{\|\mu_c - \mu_G\|_2\right\}}, \tag{2}$$

where $\text{std}$ and $\text{avg}$ represent the standard deviation and average operators, respectively. The second property, maximum equiangularity, is verified through:

$$\text{Equiangularity}_{\text{class-means}} = \text{Avg}_{c,c'}\left|\frac{\langle\mu_c - \mu_G, \mu_{c'} - \mu_G\rangle + \frac{1}{C-1}}{\|\mu_c - \mu_G\|_2\|\mu_{c'} - \mu_G\|_2}\right| \tag{3}$$

As training progresses, if the average of all class means is approaching zero, this indicates that equiangularity is being achieved.

## 4 OUT OF DISTRIBUTION NEURAL COLLAPSE

**Neural collapse in the presence of OOD data.** NC has traditionally been studied in the context of ID scenarios. However, recent empirical findings, as demonstrated in Haas et al. (2023) have shown that NC can also have a positive impact on OOD detection, especially for Lipschitz DNNs (Virmaux & Scaman, 2018). It has come to our attention that NC can influence OOD behavior as well, leading us to introduce a new property: **(NC5) ID/OOD orthogonality:** This property suggests that, as training progresses, each of the vectors representing the empirical ID class mean tends to become orthogonal to the vector representing the empirical OOD data global mean. In mathematical terms, we express this property as follows:

$$\forall c, \; \frac{\langle\mu_c, \mu_G^{\text{OOD}}\rangle}{\|\mu_c\|_2\|\mu_G^{\text{OOD}}\|_2} \to 0 \tag{4}$$

To support this observation, we examined the following metric:

$$\text{OrthoDev}_{classes-OOD} = \text{Avg}_c\left|\frac{\langle\mu_c, \mu_G^{\text{OOD}}\rangle}{\|\mu_c\|_2\|\mu_G^{\text{OOD}}\|_2}\right| \tag{5}$$

This metric assesses the deviation from orthogonality between the ID class means and the OOD mean. As training progresses, this deviation decreases towards zero if NC5 is satisfied. To validate this hypothesis, we conducted experiments using CIFAR-10 as the ID dataset and CIFAR-100 alongside SVHN (Netzer et al., 2011) as OOD datasets. We employed two different architectures, ResNet-18 (He et al., 2015) and ViT (Dosovitskiy et al., 2020), and trained each of them for 350 epochs and 6000 steps (with a batch size of 128), respectively. During training, we saved network parameters at regular intervals and evaluated the metric of equation 5 for each saved network. The results of these experiments on NC5 ( eq. 5) are shown in Figure 1, which illustrates the convergence of the OrthoDev. Both of these models were trained using ID data and were later subjected to evaluation in the presence of out-of-distribution (OOD) data. Remarkably, these models exhibited a convergence pattern characterized by a tendency to maximize their orthogonality with OOD data. This observation is substantiated by the consistently low values observed in the orthogonality equation 5. It implies that during the training process, OOD data progressively becomes more orthogonal to ID data during the TPT. In D, we present additional experiments conducted on different combinations of ID and OOD datasets, and in C.5 we give further details on NECO. We find this phenomenon intriguing and believe it can serve as the basis for a new OOD detection criterion, which we will introduce in the next Subsection.

### 4.1 NEURAL COLLAPSE BASED OUT OF DISTRIBUTION DETECTION (NECO) METHOD

Based on this observation of orthogonality between ID and OOD samples, previous works (Wang et al., 2022; Cook et al., 2020) have utilized the null space for performing OOD detection. We introduce notations and details of these methods. Given an image $x$ represented by feature vector $h_{\boldsymbol{\omega}}(x)$, we impose that $f_{\boldsymbol{\omega}}(\boldsymbol{x}) = W \times h_{\boldsymbol{\omega}}(\boldsymbol{x})$, where $W$ is the matrix of the last fully connected layer. Cook et al. (2020); Wang et al. (2022) have highlighted that features $h_{\boldsymbol{\omega}}(\boldsymbol{x})$ can

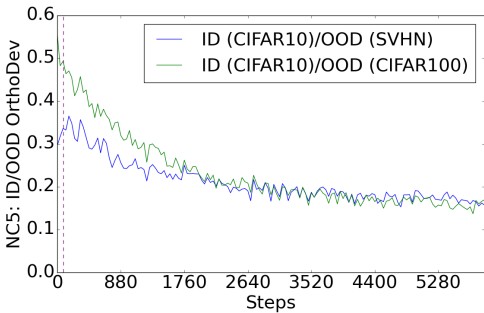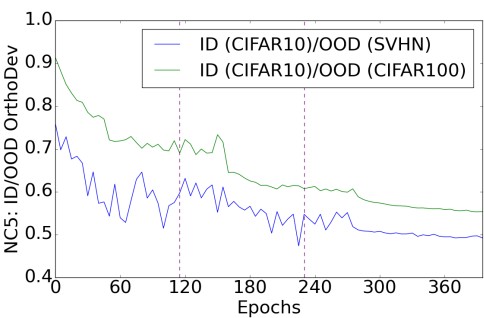

Figure 1: Convergence to ID/OOD orthogonality for ViT-B (left), Resnet-18 (right) both trained on CIFAR-10 as ID and tested in the presence of OOD data. Dashed purple lines indicate the end of warm-up steps in the case of ViT and learning rate decay epochs for ResNet-18.

be decomposed into two components: $h_{\boldsymbol{\omega}}(\boldsymbol{x}) = h_{\boldsymbol{\omega}}(\boldsymbol{x})^W + h_{\boldsymbol{\omega}}(\boldsymbol{x})^{W^\perp}$. In this decomposition, $f_{\boldsymbol{\omega}}(\boldsymbol{x}) = W \times h_{\boldsymbol{\omega}}(\boldsymbol{x})^W$, and importantly, $W \times h_{\boldsymbol{\omega}}(\boldsymbol{x})^{W^\perp} = 0$. Hence, the component $h_{\boldsymbol{\omega}}(\boldsymbol{x})^{W^\perp}$ does not directly impact classification but plays a role in influencing OOD detection. In light of this observation, NuSA introduced the following score: $\text{NuSA}(\boldsymbol{x}) = \frac{\sqrt{\|h_{\boldsymbol{\omega}}(\boldsymbol{x})\| - \|h_{\boldsymbol{\omega}}(\boldsymbol{x})^{W^\perp}\|}}{\|h_{\boldsymbol{\omega}}(\boldsymbol{x})\|}$, and ViM introduced $\text{ViM}(\boldsymbol{x}) = \|h_{\boldsymbol{\omega}}(\boldsymbol{x})^{W^\perp}\|$. To identify the null space, NuSA optimizes the decomposition after training. On the other hand, ViM conducts PCA on the latent space $H$ (also after training) and decomposes it into a principal space, defined by the $d$-dimensional projection with the matrix $P \in M_{d,D}(\mathbb{R})$ spanned by the $d$ eigenvectors corresponding to the largest $d$ eigenvalues of the co-variance matrix of $H$, and a null space, obtained by projecting on the remaining eigenvectors. In contrast, based on the implications of NC5, we propose a novel criterion that circumvents having to find the null space, specifically:

$$\text{NECO}(\boldsymbol{x}) = \frac{\|P h_{\boldsymbol{\omega}}(\boldsymbol{x})\|}{\|h_{\boldsymbol{\omega}}(\boldsymbol{x})\|} = \frac{\sqrt{h_{\boldsymbol{\omega}}(\boldsymbol{x})^\top P P^\top h_{\boldsymbol{\omega}}(\boldsymbol{x})}}{\sqrt{h_{\boldsymbol{\omega}}(\boldsymbol{x})^\top h_{\boldsymbol{\omega}}(\boldsymbol{x})}} \quad (6)$$

Our hypothesis is that if the DNN is subject to the properties of NC1, NC2, and NC5, then it should be possible to separate any subset of ID classes from the OOD data. By projecting the feature into the first $d$ principal components extracted from the ID data, we should obtain a latent space representation that is close to the null vector for OOD data and not null for ID data. Consequently, by taking the norm of this projection and normalizing it with the norm of the data, we can derive an effective OOD detection criterion. However, in the case of vision transformers, the penultimate layer representation cannot straightforwardly be interpreted as features. Consequently, the resulting norm needs to be calibrated in order to serve as a proper OOD scoring function. To calibrate our NECO score, we multiply it by the penultimate layer biggest logit (MaxLogit). This has the effect of injecting class-based information into the score in addition to the desired scaling. It is worth noting that this scaling is also useful when the penultimate layer size is smaller than the number of classes, since in this case, it is not feasible to obtain maximum OrthoDev between all classes. We refer the reader to E for empirical observations on the distribution of NECO under the presence of OOD data.

## 4.2 THEORETICAL JUSTIFICATION

To gain a better understanding of the impact of NECO, we visualize the Principal Component Analysis (PCA) of the pre-logit space in Figure 2, with ID data (colored points) and the OOD (black). This analysis is conducted on the ViT model trained on CIFAR-10, focusing on the first two principal components. Notably, the ID data points have multiple clusters, one by class, which can be attributed to the influence of NC1, and the OOD data points are positioned close to the null vector. To elucidate our criterion, we introduce an important property:

**Theorem 4.1** (NC1+NC2+NC5 imply NECO). *We consider two datasets living in $\mathbb{R}^D$, $\{D_{OOD}, D_\tau\}$ and a DNN $f_{\boldsymbol{\omega}}(\cdot) = (g_{\boldsymbol{\omega}} \circ h_{\boldsymbol{\omega}})(\cdot)$ that satisfy NC1, NC2 and NC5. There $\exists\, d \ll D$ for PCA on $D_\tau$ s.t. $NECO(\mu_G^{OOD}) = 0$. Conversely, for $\boldsymbol{x} \in D_\tau$ and considering $\boldsymbol{x} \neq \vec{0}$ we have that $NECO(\boldsymbol{x}) \neq 0$.*

The proof for this theorem can be found in A

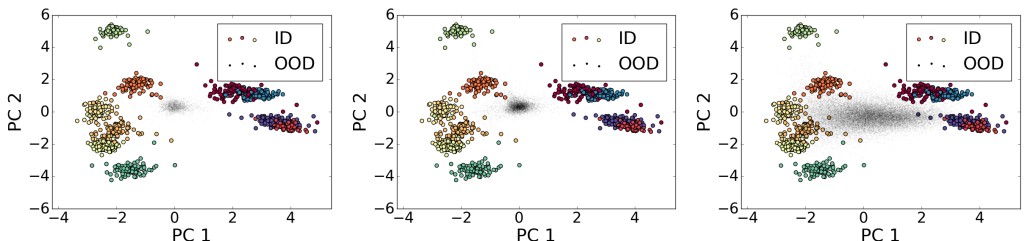

Figure 2: Feature projections on the first 2 principal components of a PCA fitted on CIFAR-10 (ID) using ViT penultimate layer representation. OOD data are ImageNet-O (left), Textures (middle), and SVHN (right). The Figure shows how NC1 (1) property is satisfied by ID data, and that OOD data lie around the origin.

**Remark 4.1.** *We have established that $NECO(\mu_G^{OOD}) = 0$, which does not imply that $NECO(\boldsymbol{x}) = 0$ for all $\boldsymbol{x} \in D_{OOD}$. However, in addition to NC5, we can put forth the hypothesis that NC2 also occurs on the mix of ID/OOD data, while NC1 doesn't occur on the OOD data by themselves, resulting in an OOD data cluster that is equiangular and orthogonal to the Simplex ETF structure. We refer the reader to Section D for justification. Furthermore, the observations made in Figure 2 appear to support our hypothesis.*

## 5 EXPERIMENTS & RESULTS

In this section, we compare our method with state-of-the-art algorithms, on small-scale and large-scale OOD detection benchmarks. Following prior work, we consider ImageNet-1K, CIFAR-10, and CIFAR-100 as the ID datasets. We use both Transformer-based and CNN-based models to benchmark our method. Detailed experimental settings are as follows.

**OOD Datasets.** For experiments involving ImageNet-1K as the inliers dataset (ID), we assess the model's performance on five OOD benchmark datasets: Textures (Cimpoi et al., 2014), Places365 (Zhou et al., 2016), iNaturalist (Horn et al., 2017), a subset of 10 000 images sourced from (Huang & Li, 2021a), ImageNet-O (Hendrycks et al., 2021b) and SUN (Xiao et al., 2010). For experiments where CIFAR-10 (resp. CIFAR-100) serves as the ID dataset, we employ CIFAR-100 (resp. CIFAR-10) alongside the SVHN dataset (Netzer et al., 2011) as OOD datasets in these experiments. The standard dataset splits, featuring 50 000 training images and 10 000 test images, are used in these evaluations. Further details are provided in C.1. We refer the reader to E for additional testing results on the OpenOOD benchmark (Zhang et al., 2023).

**Evaluation Metrics.** We present our results using two widely adopted OOD metrics (Yang et al., 2021). Firstly, we consider FPR95, the false positive rate when the true positive rate (TPR) reaches 95%, with smaller values indicating superior performance. Secondly, we use the AUROC metric, which is threshold-free and calculates the area under the receiver operating characteristic curve (TPR). A higher value here indicates better performance. Both metrics are reported as percentages.

**Experiment Details.** We evaluate our method on a variety of neural network architectures, including Transformers and CNNs. ViT (Vision Transformer) (Dosovitskiy et al., 2020) is a Transformer-based image classification model which treats images as sequences of patches. We take the official pretrained weights on ImageNet-21K (Dosovitskiy et al., 2020) and fine-tune them on ImageNet-1K, CIFAR-10, and CIFAR-100. Swin (Liu et al., 2021) is also a transformer-based classification model. We use the officially released SwinV2-B/16 model, which is pre-trained on ImageNet-21K and fine-tuned on ImageNet-1K. From the realm of CNN-based models, we use ResNet-18 (He et al., 2015). When estimating the simplex ETF space, the entire training set is used. Further details pertaining to model training are provided in C.4. As a complementary experiment, we also assess our approach on DeiT Touvron et al. (2021) in E

**Baseline Methods.** In our evaluation, we compared our method against twelve prominent post-hoc baseline approaches, which we list and provide ample details pertaining to the implementation and associated hyper-parameters in B.

| Model | Method | ImageNet-O | | Textures | | iNaturalist | | SUN | | Places365 | | Average | |
|---|---|---|---|---|---|---|---|---|---|---|---|---|---|
| | | AUROC↑ | FPR↓ | AUROC↑ | FPR↓ | AUROC↑ | FPR↓ | AUROC↑ | FPR↓ | AUROC↑ | FPR↓ | AUROC↑ | FPR↓ |
| ViT-B/16 | Softmax score | 85.31 | 52.65 | 86.64 | 49.40 | 97.21 | 12.14 | 86.64 | 53.75 | 85.52 | 56.93 | 88.26 | 44.97 |
| | MaxLogit | 92.23 | 36.40 | 91.69 | 36.90 | 98.87 | 5.72 | 92.15 | 39.88 | 90.05 | 46.75 | 92.99 | 33.13 |
| | Energy | 93.03 | 31.65 | 92.13 | 34.15 | 99.04 | 4.76 | 92.65 | 36.90 | 90.38 | 43.72 | 93.45 | 30.24 |
| | Energy+ReAct | 93.08 | 31.25 | 92.08 | 34.50 | 99.02 | 4.91 | 92.56 | 36.94 | 90.19 | 44.25 | 93.39 | 30.37 |
| | ViM | 94.01 | 28.95 | 91.63 | 38.22 | 99.59 | 2.00 | 92.73 | 34.00 | 89.67 | 44.12 | 93.56 | 29.46 |
| | Residual | 92.10 | 38.35 | 87.71 | 52.34 | 99.36 | 2.79 | 89.46 | 40.91 | 85.60 | 53.93 | 90.85 | 37.66 |
| | GradNorm | 84.70 | 34.85 | 86.29 | 35.76 | 97.45 | 6.17 | 86.67 | 39.49 | 83.59 | 46.94 | 87.74 | 32.64 |
| | Mahalanobis | 94.00 | 30.90 | 91.69 | 37.93 | 99.67 | 1.55 | 91.38 | 38.48 | 88.66 | 46.09 | 93.08 | 30.99 |
| | KL-Matching | 82.90 | 53.40 | 84.61 | 52.38 | 95.07 | 15.31 | 83.25 | 62.10 | 82.00 | 65.99 | 85.57 | 49.84 |
| | ASH-B | 69.28 | 85.25 | 66.05 | 83.29 | 72.62 | 80.62 | 59.36 | 91.53 | 55.08 | 92.68 | 64.48 | 86.67 |
| | ASH-P | 93.31 | 29.00 | 91.55 | 37.58 | 98.75 | 6.25 | 92.96 | 33.50 | 90.07 | 41.81 | 93.33 | 29.63 |
| | ASH-S | 92.85 | 29.45 | 91.00 | 39.73 | 98.50 | 7.28 | 92.61 | 33.12 | 89.55 | 41.68 | 92.90 | 30.25 |
| | NuSA | 92.48 | 34.30 | 88.28 | 51.24 | 99.30 | 3.12 | 89.26 | 40.08 | 85.28 | 54.51 | 90.92 | 36.65 |
| | **NECO (ours)** | **94.53** | **25.20** | **92.86** | **32.44** | 99.34 | 3.26 | **93.15** | **33.98** | **90.38** | 42.66 | **94.05** | **27.51** |
| SwinV2 | Softmax score | 61.09 | 89.60 | 81.72 | 60.91 | 88.59 | 47.66 | 81.24 | 66.85 | 81.09 | 68.27 | 78.75 | 66.66 |
| | MaxLogit | 61.34 | 87.95 | 80.36 | 59.55 | 86.47 | 50.51 | 78.17 | 68.50 | 77.43 | 69.41 | 76.75 | 67.18 |
| | Energy | 61.32 | 85.75 | 77.91 | 64.44 | 81.85 | 63.44 | 73.80 | 76.89 | 73.09 | 76.68 | 73.59 | 73.44 |
| | Energy+ReAct | 68.38 | 83.85 | **84.56** | 59.86 | 90.23 | 51.97 | **82.61** | 69.27 | 81.41 | 70.19 | **81.44** | 67.03 |
| | ViM | 69.06 | 83.45 | 81.50 | 61.18 | 87.54 | 54.09 | 75.24 | 73.92 | 73.12 | 76.46 | 77.29 | 69.82 |
| | Residual | 66.82 | 83.80 | 77.36 | 65.00 | 83.23 | 59.77 | 71.03 | 76.41 | 68.90 | 78.53 | 73.47 | 72.70 |
| | GradNorm | 37.39 | 95.95 | 33.84 | 93.31 | 31.82 | 95.01 | 31.97 | 96.29 | 33.06 | 95.73 | 33.62 | 95.26 |
| | Mahalanobis | **71.87** | 86.05 | 84.51 | 63.35 | 89.81 | 57.10 | 80.28 | 75.39 | 78.52 | 77.10 | 80.99 | 71.80 |
| | KL-Matching | 58.60 | 87.50 | 75.30 | 71.69 | 82.93 | 58.29 | 73.72 | 76.53 | 72.11 | 78.91 | 72.53 | 74.58 |
| | ASH-B | 47.07 | 96.35 | 38.59 | 97.50 | 48.62 | 97.55 | 52.11 | 95.64 | 52.93 | 96.18 | 47.86 | 96.64 |
| | ASH-P | 37.38 | 97.95 | 26.51 | 98.90 | 20.73 | 99.28 | 24.49 | 99.08 | 26.12 | 98.93 | 27.05 | 98.83 |
| | ASH-S | 40.36 | 95.10 | 36.08 | 94.63 | 16.15 | 99.50 | 22.21 | 98.53 | 23.72 | 98.05 | 27.70 | 97.16 |
| | NuSA | 56.50 | 91.95 | 62.72 | 83.14 | 64.01 | 83.58 | 55.97 | 91.28 | 54.44 | 92.71 | 58.73 | 88.53 |
| | **NECO (ours)** | 65.03 | **80.55** | 82.27 | **54.67** | **91.89** | **34.41** | 82.13 | **62.26** | 81.46 | **64.08** | 80.56 | **59.19** |

Table 1: OOD detection for NECO and baseline methods. The ID dataset is ImageNet-1K, and OOD datasets are Textures, ImageNet-O, iNaturalist, SUN, and Places365. Both metrics AUROC and FPR95. are in percentage. A pre-trained ViT-B/16 model and SwinV2 are tested. The best method is emphasized in bold, and the 2nd and 3rd ones are underlined.

**Result on ImageNet-1K.** In Table 1 we present the results for the ViT model in the first half. The best AUROC values are highlighted in bold, while the second and third best results are underlined. Our method demonstrates superior performance compared to the baseline methods. Across four datasets, we achieve the highest average AUROC and the lowest FPR95. Specifically, we attain a FPR95 of 27.51%, surpassing the second-place method, ViM, by 1.95%. The only dataset where we fall slightly behind is iNaturalist, with our AUROC being only 0.33% lower than the best-performing approach. In Table 1, we provide the results for the SwinV2 model in the second half. Notably, our method consistently outperforms all other methods in terms of FPR95 on all datasets. On average, we achieve the third-best AUROC and significantly reduce the FPR95 by 7.47% compared to the second-best performing method, Softmax score. We compare NECO with MaxLogit (Hendrycks et al., 2022) to illustrate the direct advantages of our scoring function. On average, we achieve a FPR95 reduction of 5.62% for ViT and 7.99% for Swin when transitioning from MaxLogit to NECO multiplied by the MaxLogit. This performance enhancement clearly underscores the value of incorporating the NC concept into the OOD detection framework. Moreover, NECO is straightforward to implement in practice, requiring simple post-hoc linear transformations and weight masking.

**Results on CIFAR.** Table 2 presents the results for the ViT model and Resnet-18 both on CIFAR-10 and CIFAR-100 as ID dataset, tested against different OOD datasets. On the first half of the table, we show the results using a ViT model. On the majority of the OOD datasets cases we outperform the baselines both in terms of FPR95 and AUROC. Only ASH outperforms NECO on the use case CIFAR-100 vs SVHN. On average we surpass all the baseline on both test sets. In the second half, we show the results using a ResNet-18 model. Similarly to ViT results, on average we surpass the baseline strategies by at least 1.28% for the AUROC on the CIFAR-10 cases and lowered the best baseline performance by 8.67% in terms of FPR95, on the CIFAR-100 cases. On average, our

| | | CIFAR-100 | | SVHN | | Average | | CIFAR-10 | | SVHN | | Average | |
|---|---|---|---|---|---|---|---|---|---|---|---|---|---|
| | | AUROC↑ | FPR↓ | AUROC↑ | FPR↓ | AUROC↑ | FPR↓ | AUROC↑ | FPR↓ | AUROC↑ | FPR↓ | AUROC↑ | FPR↓ |
| ViT | Softmax score | 98.36 | 7.40 | 99.64 | 0.59 | 99.00 | 3.99 | 92.13 | 34.74 | 91.29 | 39.56 | 91.71 | 37.15 |
| | MaxLogit | 98.60 | 5.94 | 99.90 | 0.23 | 99.25 | 3.09 | 92.81 | 26.73 | 96.20 | 18.68 | 94.51 | 22.71 |
| | Energy | 98.63 | 5.93 | 99.92 | 0.21 | 99.28 | 3.07 | 92.68 | 26.37 | 96.68 | 15.72 | 94.68 | 21.05 |
| | Energy+ReAct | 98.68 | 5.92 | 99.92 | 0.23 | 99.30 | 3.08 | 93.43 | 25.93 | 96.65 | 15.88 | 95.04 | 20.91 |
| | ViM | 98.81 | 4.86 | 99.50 | 0.82 | 99.16 | 2.84 | 94.78 | 26.10 | 95.55 | 26.33 | 95.17 | 26.22 |
| | Residual | 98.49 | 7.09 | 96.95 | 12.64 | 97.72 | 9.87 | 94.63 | 28.42 | 92.43 | 44.65 | 93.53 | 36.54 |
| | GradNorm | 95.82 | 10.92 | 99.85 | 0.48 | 97.84 | 5.7 | 92.10 | 26.20 | 94.85 | 16.69 | 93.48 | 21.45 |
| | Mahalanobis | 98.73 | 5.95 | 95.52 | 18.61 | 97.13 | 12.28 | 95.42 | 24.02 | 93.91 | 37.11 | 94.67 | 30.57 |
| | KL-Matching | 83.48 | 20.65 | 95.07 | 6.09 | 89.28 | 13.37 | 79.47 | 38.34 | 80.19 | 41.73 | 79.83 | 40.04 |
| | ASH-B | 95.15 | 17.49 | 99.09 | 4.81 | 97.12 | 11.15 | 78.07 | 53.01 | 82.99 | 53.97 | 80.53 | 53.49 |
| | ASH-P | 97.89 | 7.22 | 99.90 | 0.26 | 98.90 | 3.74 | 85.12 | 29.82 | 97.04 | 14.06 | 91.08 | 21.94 |
| | ASH-S | 97.56 | 7.73 | 99.89 | 0.32 | 98.73 | 4.03 | 83.30 | 30.96 | 97.04 | 14.55 | 90.17 | 22.56 |
| | NuSA | 98.56 | 6.79 | 99.60 | 1.03 | 99.08 | 3.91 | 94.50 | 28.77 | 94.27 | 35.91 | 94.39 | 32.34 |
| | **NECO (ours)** | **98.95** | **4.81** | **99.93** | **0.12** | **99.44** | **2.47** | 95.17 | 23.39 | 96.65 | 15.47 | **95.41** | **19.43** |
| ResNet-18 | Softmax score | 85.07 | 67.41 | 92.14 | 46.77 | 88.61 | 57.09 | 75.35 | 83.09 | 77.30 | 85.61 | 76.33 | 84.35 |
| | MaxLogit | 85.36 | 59.19 | 93.73 | 31.52 | 89.55 | 45.36 | 75.44 | 83.12 | 77.24 | 87.59 | 76.34 | 85.36 |
| | Energy | 85.46 | 58.68 | 93.96 | 29.71 | 89.71 | 44.20 | 75.25 | 83.76 | 76.40 | 90.77 | 75.83 | 87.27 |
| | Energy+ReAct | 84.22 | 60.04 | 92.31 | 35.17 | 88.27 | 47.61 | 70.00 | 85.99 | 74.14 | 91.05 | 72.07 | 88.52 |
| | ViM | 85.11 | 63.76 | 94.87 | 27.35 | 89.99 | 45.56 | 67.61 | 90.08 | 86.13 | 67.31 | 76.87 | 78.70 |
| | Residual | 76.06 | 76.53 | 90.21 | 48.18 | 83.14 | 62.36 | 45.99 | 96.19 | 72.52 | 86.60 | 59.23 | 91.40 |
| | GradNorm | 60.83 | 69.98 | 78.28 | 42.83 | 69.56 | 56.41 | 72.62 | 84.64 | 69.66 | 91.58 | 59.55 | 88.11 |
| | Mahalanobis | 81.23 | 72.35 | 90.39 | 54.51 | 85.81 | 63.43 | 55.85 | 95.41 | 79.96 | 81.63 | 67.91 | 88.52 |
| | KL-Matching | 77.83 | 68.12 | 86.73 | 49.47 | 82.28 | 58.80 | 73.52 | 83.05 | 77.32 | 74.82 | 67.91 | 78.94 |
| | ASH-B | 73.89 | 64.45 | 83.79 | 44.77 | 78.84 | 54.61 | 74.10 | 84.00 | 78.14 | 82.02 | 76.12 | 83.01 |
| | ASH-P | 83.96 | 59.90 | 94.77 | 26.04 | 89.37 | 42.97 | **75.46** | **82.98** | 77.28 | 89.43 | 76.37 | 86.21 |
| | ASH-S | 82.48 | 62.38 | 91.54 | 39.94 | 87.01 | 51.16 | 72.17 | 85.20 | 80.18 | 74.39 | 76.18 | 80.00 |
| | NuSA | 85.65 | 59.52 | 95.77 | 21.82 | 90.71 | 40.67 | 70.32 | 86.36 | 88.44 | 57.58 | 79.38 | 71.97 |
| | **NECO (ours)** | **86.61** | **57.81** | **95.92** | **20.43** | **91.27** | **39.12** | 70.26 | 85.70 | **88.57** | 54.35 | **79.42** | **70.03** |

Table 2: OOD detection for NECO vs baseline methods. The ID dataset are CIFAR-10/CIFAR-100, and OOD datasets are CIFAR-100/CIFAR-10 alongside SVHN. Both metrics AUROC and FPR95 are in percentage. The best method is emphasized in bold, and the 2nd and 3rd ones are underlined.

approach outperforms baseline methods in terms of AUROC. However, we notice that our method performs slightly worse on the CIFAR-100-ID/CIFAR-10-OOD task.

# 6 CONCLUSION

This paper introduces a novel OOD detection method that capitalizes on the Neural Collapse (NC) properties inherent in DNNs. Our empirical findings demonstrate that when combined with over-parameterized DNNs, our post-hoc approach, NECO, achieves state-of-the-art OOD detection results, surpassing the performance of most recent methods on standard OOD benchmark datasets. While many existing approaches focus on modeling the noise by either clipping it or considering its norm, NECO takes a different approach by leveraging the prevalent ID information in the DNN, which will only be enhanced with model improvements. The latent space contains valuable information for identifying outliers, and NECO incorporates an orthogonal decomposition method that preserves the equiangularity properties associated with NC.

We have introduced a novel NC property (NC5) that characterizes both ID and OOD data behavior. Through our experiments, we have shed light on the substantial impact of NC on the OOD detection abilities of DNNs, especially when dealing with over-parameterized models. We observed that NC properties, including NC5, NC1, and NC2, tend to converge towards zero as expected when the network exhibits over-parameterization. This empirical observation provides insights into why our approach performs exceptionally on a variety of models and on a broad range of OOD datasets (we refer the reader to the complementary results in E), hence demonstrating the robustness of NECO against OOD data. While our approach excels with Transformers and CNNs, especially in the case of over-parametrized models, we observed certain limitations when applying it to DeiT (see the results in E). These limitations may be attributed to the distinctive training process of DeiT (*i.e.*, including a distillation strategy) which necessitates a specific setup that we did not account for in order to prevent introducing bias to the NC phenomenon. We hope that this work has shed some light on the interactions between NC and OOD detection. Finally, based on our results and observations, this work raises new research questions on the training strategy of DNNs that lead to NC in favor of OOD detection.

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

TABLE OF CONTENTS - SUPPLEMENTARY MATERIAL

## A  THEORETICAL JUSTIFICATION

Before proceeding with the proof of Theorem 4.1, we first introduce some notation and a useful lemma for the subsequent proof.

Given a vector $z_1 \in \mathbb{R}^D$ we define $z_1^{PCA}$ as the projection of $z_1$ onto a subspace obtained by applying PCA to an external dataset $D$ of same dimensionality as $z_1$. This subspace has dimensions limited to $d \leq D$.

**Lemma A.1** (Orthogonality conservation). *We consider a single dataset, $D_1 = \{h_i\}_{i=1}^n \in \mathbb{R}^D$, along with two vectors, $\{z_1, z_2\} \in \mathbb{R}^D$, that are not necessarily part of $D_1$ with the condition that $\langle z_1, z_2 \rangle = 0$. Considering a PCA on $D_1$, it follows that $\langle z_1^{PCA}, z_2^{PCA} \rangle = 0$.*

*Proof.* If $z_1 \perp z_2$, then we have $z_1^\top z_2 = 0$. Let us define $P \in \mathbb{M}_{D,d}(\mathbb{R})$ the projection matrix of the $PCA$.
By definition $P$ is an orthogonal matrix, hence $P^\top P = I_d$ and $PP^\top = I_D$, where $I_D$ and $I_d$ are the identity matrices of $\mathbb{M}_{D \times D}(\mathbb{R})$ and $\mathbb{M}_{d \times d}(\mathbb{R})$, respectively. We have $(z_1^{PCA})^\top z_2^{PCA} = (P^\top z_1)^\top (P^\top z_2) = z_1^\top PP^\top z_2 = z_1^\top I_D z_2 = z_1^\top z_2 = 0$, thus proving the orthogonality of PCA projections. $\square$

This lemma asserts that the PCA projection keeps the orthogonal property of two vectors, which will be of great importance towards the proof of the next theorem.

For completeness, we reiterate the theorem, followed by the proof.

**Theorem A.2** (NC1+NC2+NC5 imply NECO). *We consider two datasets living in $\mathbb{R}^D$, $\{D_{OOD}, D_\tau\}$ and a DNN $f_\omega(\cdot) = (g_\omega \circ h_\omega)(\cdot)$ that satisfy NC1, NC2 and NC5. There $\exists\, d \ll D$ for PCA on $D_\tau$ s.t. $NECO(\mu_G^{OOD}) = 0$. Conversely, for $x \in D_\tau$ and considering $x \neq \vec{0}$ we have that $NECO(x) \neq 0$.*

*Proof.* Assuming, for the sake of contradiction, that we have $\forall d \leq D$ s.t. $\text{NECO}(\mu_G^{\text{OOD}}) \neq 0$ and $\text{NECO}(x) = 0$ for data points $x$ in $D_\tau$. Given a PCA dimension $d$ s.t. $d \geq C$ and based on the mathematical underpinnings of PCA (*i.e.*, linear nature of projections) coupled with the NC1+NC2 assumptions, most data points $x$ in $D_\tau$ satisfy the condition that $\|Ph_\omega(x)\|$ is maximized and therefore not equal to zero, implicitly the NECO score not being equal to zero.
Considering our assumption of NC1 + NC2, we can infer that each class $c$ resides within a one-dimensional space that is aligned with $\mu_c$. As a result, the set of vectors $\{\mu_c\}_{c=1}^C$ forms an orthogonal basis. According to the PCA definition, the dimensional space resulting from PCA, when $d = C$, should be spanned by the set $\{\mu_c^{\text{PCA}}\}_{c=1}^C$. Here, for each class $c$, $\mu_c^{\text{PCA}}$ represents the PCA projection of $\mu_c$.
Utilizing the orthogonality conservation lemma and considering our assumption of NC5, we can deduce that each $\mu_c^{\text{PCA}}$ is orthogonal to $\mu_G^{\text{OOD}}$. Consequently, we can confidently state that $\text{NECO}(\mu_G^{\text{OOD}}) = 0$, thereby achieving the sought contradiction. $\square$

## B  DETAILS ON BASELINES IMPLEMENTATIONS

In this section, we provide an overview of the various baseline methods utilized in our experiments. We explain the mechanisms underlying these baselines, detail the hyperparameters employed, and offer insights into the process of determining hyperparameters when a baseline was not originally designed for a specific architecture.

**ASH.**    ASH, as introduced by Djurisic et al. (2022), employs activation pruning at the penultimate layer, just before the application of the DNN classifier. This pruning threshold is determined on a per-sample basis, eliminating the need for pre-computation of ID data statistics. The original paper presents three different post-hoc scoring functions, with the only distinction among them being the imputation method applied after pruning. In ASH-P, the clipped values are replaced with zeros. ASH-B substitutes them with a constant equal to the sum of the feature vector before pruning, divided by the number of kept features. ASH-S imputes the pruned values by taking

the exponential of the division between the sum of the feature vector before pruning and the feature vector after pruning. Since our study employs different models than the original ASH paper by Djurisic et al. (2022), we fine-tuned their method using a range of threshold values, specifically $[65, 70, 75, 80, 85, 90, 95, 99]$, for their three variations of ASH, consistent with the thresholds used in the original paper. The optimized ASH thresholds are presented in Table B.3.

| Ash Variant | ID dataset / Model | CIFAR-10 | CIFAR-100 | ImageNet |
|---|---|---|---|---|
| ASH-B | ViT | 75 | 60 | 70 |
|  | SwinV2-B/16 | - | - | 99 |
|  | DeiT-B/16 | - | - | 60 |
|  | ResNet-18 | 75 | 90 | - |
| ASH-P | ViT | 60 | 60 | 60 |
|  | SwinV2-B/16 | - | - | 60 |
|  | DeiT-B/16 | - | - | 60 |
|  | ResNet-18 | 95 | 85 | - |
| ASH-S | ViT | 60 | 60 | 60 |
|  | SwinV2-B/16 | - | - | 99 |
|  | DeiT-B/16 | - | - | 99 |
|  | ResNet-18 | 60 | 85 | - |

Table B.3: Ash optimised pruning threshold per-model and per-dataset. Values are in percentage.

**ReAct.** For the ReAct baseline, we adopt the ReAct+Energy setting, which has been identified as the most effective configuration in prior work Sun et al. (2021). The original paper determined that the 90th percentile of activations estimated from the in-distribution (ID) data was the optimal threshold for clipping. However, given that we employed different models in our experiments, we found that using a rectification percentile of $p = 99$ yielded better results in terms of reducing the false positive rate (FPR95) for models like ViT and DeiT, while a percentile of $p = 0.95$ was more suitable for Swin. In our reported results, we use the best threshold setting for each model accordingly to ensure optimal performance.

**ViM and Residual.** Introduced by Wang et al. (2022), ViM decomposes the latent space into a principal space $P$ and a null space $P^{\perp}$. The ViM score is then computed by using the projection of the features on the null space to create a virtual logit, using the norm of this projection along with the logits. In addition, they calibrate the obtained norm by a constant alpha to enhance their performance. Alpha is the division of the sum of the Maximum logits, divided by the sum of the norms in the null space projection, both measured on the training set.
As for Residual, the score is obtained by computing the norm of the latent vector projection in the null space.
We followed ViM official paper (Wang et al., 2022) recommendation for the null space dimenssion in the ViT, DeiT and swin, i.e., [512,512,1000] respectively. However, the case of ResNet-18 was not considered in the paper. We found that a 300 latent space dimension works best in reducing the FPR.

**Mahalanobis.** We have also implemented the Mahalanobis score by utilizing the feature vector before the final classification layer, which corresponds to the DNN classifier, following the methodology outlined in Fort et al. (2021). To estimate the precision matrix and the class-wise average vector, we utilize the complete training dataset. It is important to note that we incorporate the ground-truth labels during the computation process.

**KL-Matching.** We calculate the class-wise average probability by considering the entire training dataset. In line with the approach described in Hendrycks et al. (2022), we rely on the predicted class rather than the ground-truth labels for this calculation.

**Softmax score.** Hendrycks & Gimpel (2017) uses the maximum softmax probability (MSP) of the model as an OOD scoring function.

**MaxLogit.** Hendrycks et al. (2022) uses the maximum logit value (Maxlogit) of the model as an OOD scoring function.

**GradNorm.** Huang & Li (2021a) computes the norm of gradients between the softmax output and a uniform probability distribution.

**Energy.** Liu et al. (2020) proposes to use the energy score for OOD detection. The energy function maps the logit outputs to a scalar $Energy(x; f) \in \mathbb{R}$. In order to keep the convention that the score need to be lower for ID data, Liu et al. (2020) used the negative energy as the OOD score.

## C ADDITIONAL EXPERIMENT DETAILS

### C.1 OOD DATASETS

For experiments involving ImageNet-1K as the inliers dataset (ID), we assess the model's performance on five OOD benchmark datasets. Textures (Cimpoi et al., 2014) dataset comprises natural textural images, with the exclusion of four categories (*bubbly*, *honeycombed*, *cobwebbed*, *spiralled*) that overlap with the inliers dataset. Places365 (Zhou et al., 2016) is a dataset consisting of images depicting various scenes. iNaturalist (Horn et al., 2017) dataset focuses on fine-grained species classification. We utilize a subset of 10 000 images sourced from (Huang & Li, 2021a). ImageNet-O (Hendrycks et al., 2021b) is a dataset comprising adversarially filtered images designed to challenge OOD detectors. SUN (Xiao et al., 2010) is a dataset containing images representing different scene categories. Figure C.3 shows five examples from each of these datasets.
For experiments where CIFAR-10 (resp. CIFAR-100) serves as the ID dataset, we employ CIFAR-100 (resp. CIFAR-10) as OOD dataset. Additionally, we include the SVHN dataset (Netzer et al., 2011) as OOD datasets in these experiments. The standard dataset splits, featuring 50 000 training images and 10 000 test images, are used in these evaluations.

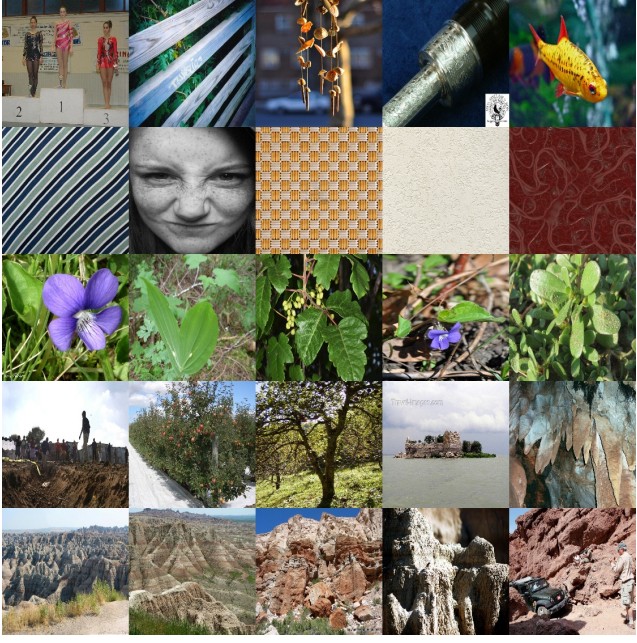

Figure C.3: Example images from ImageNet-1k considered OOD datasets. Each row representing an OOD dataset: ImageNet-O, Textures, iNaturalist, SUN and Places365 respectively from top to bottom.

### C.2 NECO PSEUDO-CODE

Algorithm 1 presents the pseudo-code of the process utilised to compute NECO during inference. This assumes that a PCA is already computed on the training data, the DNN is trained and a threshold on the score is already identified.

---

**Algorithm 1:** NECO OOD detection pseudo code

---

**Data:** $X$

;                        `// inference data samples`

**Inputs:** $model, pca, thres$ ;     `// where model is the DNN, pca is model estimated on training data`
  `using K main principal components, and thres a threshold selected after the validation with the ROC`
  `Curve`

$X_{Features} = model(X)$
$ETF_{X_{features}} = pca(X)$
$score = \dfrac{Norm(ETF_{X_{features}})}{Norm(X_{Features})}$

**if** $score > threshold$ **then**
  |   $isOOD$=false
**else**
  |   $isOOD$=true
**end**
**return** $score, isOOD$

---

## C.3   EFFECT OF MAXLOGIT MULTIPLICATION

Table C.4 shows the performance of NECO with and without the maximum logit multiplication. We observe that using maximum logit considerably improve the performance average than the standalone NECO score. This gap can be explained by the limited size of the feature space in vision transformers (*e.g.* 768 in the case of ViT) which is insufficient to converge perfectly to the Simplex ETF structure due to the number of classes within ImageNet-1k dataset. In addition, we can observe a larger performance gap for the DeiT case. We suggest that this might be due due to the specific training procedure of DeiT, relying on a distillation process, which might hinder its collapse.

| Model | Method | ImageNet-O | | Textures | | iNaturalist | | SUN | | Places365 | | Average | |
|---|---|---|---|---|---|---|---|---|---|---|---|---|---|
| | | AUROC↑ | FPR↓ | AUROC↑ | FPR↓ | AUROC↑ | FPR↓ | AUROC↑ | FPR↓ | AUROC↑ | FPR↓ | AUROC↑ | FPR↓ |
| ViT | **NECO w/o Maxlogit** | 93.27 | 30.05 | 89.02 | 46.96 | **99.48** | **1.47** | 89.90 | 38.24 | 85.89 | 52.15 | 91.51 | 33.77 |
| | **NECO** | **94.53** | **25.20** | **92.86** | **32.44** | 99.34 | 3.26 | **93.15** | **33.98** | **90.38** | **42.66** | **94.05** | **27.51** |
| SwinV2 | **NECO w/o Maxlogit** | 63.17 | 87.95 | 75.52 | 69.63 | 90.72 | 36.37 | 81.59 | **57.19** | 80.52 | **60.77** | 78.31 | 62.38 |
| | **NECO** | **65.03** | **80.55** | **82.27** | **54.67** | **91.89** | **34.41** | **82.13** | 62.26 | **81.46** | 64.04 | **80.56** | **59.19** |
| DeiT | **NECO w/o Maxlogit** | 56.21 | 95.80 | 60.02 | 96.68 | 71.41 | 84.49 | 57.29 | 84.10 | 55.54 | 82.74 | 60.10 | 88.76 |
| | **NECO** | **62.72** | **84.10** | **80.82** | **59.47** | **89.54** | **42.26** | **77.64** | **65.55** | **76.48** | **68.13** | **77.44** | **63.90** |
| ResNet 50 | **NECO w/o Maxlogit** | **71.01** | **85.44** | 84.86 | 60.79 | 81.04 | 84.85 | 60.42 | 96.77 | 61.92 | 96.34 | 71.85 | 84.83 |
| | **NECO** | 69.80 | 86.43 | **88.09** | **51.53** | **87.94** | 62.69 | **75.56** | **77.55** | **73.07** | **78.62** | **78.89** | **71.36** |

Table C.4: OOD detection performance for NECO with versus without Maxlogit multiplication. Both metrics AUROC and FPR95. are in percentage. A pre-trained ViT-B/16 is used for testing.

## C.4   FINE-TUNING DETAILS

The fine-tuning setup for the ViT model is as follows: we take the official pre-trained weights on ImageNet-21K (Dosovitskiy et al., 2020) and fine-tune them on ImageNet-1K, CIFAR-10, and CIFAR-100. For ImageNet-1K, the weights are fine-tuned for $18\,000$ steps, with 500 cosine warm-up steps, 256 batch size, 0.9 momentum, and an initial learning rate of $2\mathrm{x}10^{-2}$. For CIFAR-10 and CIFAR-100 the weights are fine-tuned for 500 and 1000 steps respectively. With 100 warm-up steps, 512 batch size, and the rest of the training parameters being equal to the case of ImageNet-1K. Swin (Liu et al., 2021) is also a transformer-based classification model. We use the officially released SwinV2-B16 model, which is pre-trained on ImageNet-21K and fine-tuned on ImageNet-1K. ResNet-18 (He et al., 2015) is a CNN-based model. For both CIFAR-10 and CIFAR-100, the model is trained for 200 epochs with 128 batch size, $5\mathrm{x}10^{-4}$ weight decay and 0.9 momentum. The initial learning rate is 0.1 with a cosine annealing scheduler. When estimating the simplex ETF space, the entire training set is used. Note That for in the imagenet case, we used an input size of 384 for ViT, while the image size was only 224 for Swin and DeiT models, which might contribute to the higher performance of the ViT model.

## C.5 DIMENSION OF EQUIANGULAR TIGHT FRAME (ETF)

In Figures C.5,C.6 and C.4, we show the performances, in terms of AUROC and FPR95, for different models, with different ID data: ImageNet, CIFAR-10 or CIFAR-100. We observe that both performance metrics remain relatively stable once the dimension rises over a certain threshold for each case study. This suggests that including more dimensions beyond this threshold adds little to no meaningful information about the ID data. We hypothesize that the inflection point in these curves indicates the best approximate subspace dimension of the Simplex ETF relevant to our ID data. Therefore, for greater adaptability, Table C.5 shows the best ETF values that minimize the FPR (and maximize the AUC in case of equal FPR) for each tuple dataset. Hence, our results depicted in Section 5 utilize the best ETF values for each tuple <model, ID, OOD>. In the more general case, according to Figures C.5,C.6 and C.4, we still have the flexibility to find a single ETF dimension per ID dataset and model if the specific application demands it while maintaining state-of-the-art performance. For the evaluation/inference phase, the threshold can be fixed at a dimension $d$ such that the first $d$ principal components that explain at least 90% of the ID variance from the train dataset.

| ID Data | Model | ImageNet-O | Textures | iNaturalist | SUN | Places365 | CIFAR-10 | CIFAR-100 | SVHN |
|---|---|---|---|---|---|---|---|---|---|
| | ViT-B/16 | 140 | 270 | 490 | 270 | 360 | - | - | - |
| ImageNet | SwinV2-B/16 | 70 | 510 | 450 | 560 | 670 | - | - | - |
| | DeiT-B/16 | 430 | 670 | 370 | 710 | 730 | - | - | - |
| CIFAR-10 | ViT-B/16 | - | - | - | - | - | - | 40 | 100 |
| | ResNet-18 | - | - | - | - | - | - | 130 | 250 |
| CIFAR-100 | ViT-B/16 | - | - | - | - | - | 130 | - | 260 |
| | ResNet-18 | - | - | - | - | - | 150 | - | 470 |

Table C.5: Best Simplex ETF approximate dimension, for different DNN architectures and ID data case studies: ImageNet, CIFAR-10, and CIFAR-100. The Best ETF dimension is found by minimizing the FPR95, then maximizing the AUROC in case of equal FPR95 across the dimensions. The values are presented for different DNN architectures and ID data (ImageNet, CIFAR-10, and CIFAR-100) and their respective OOD test cases.

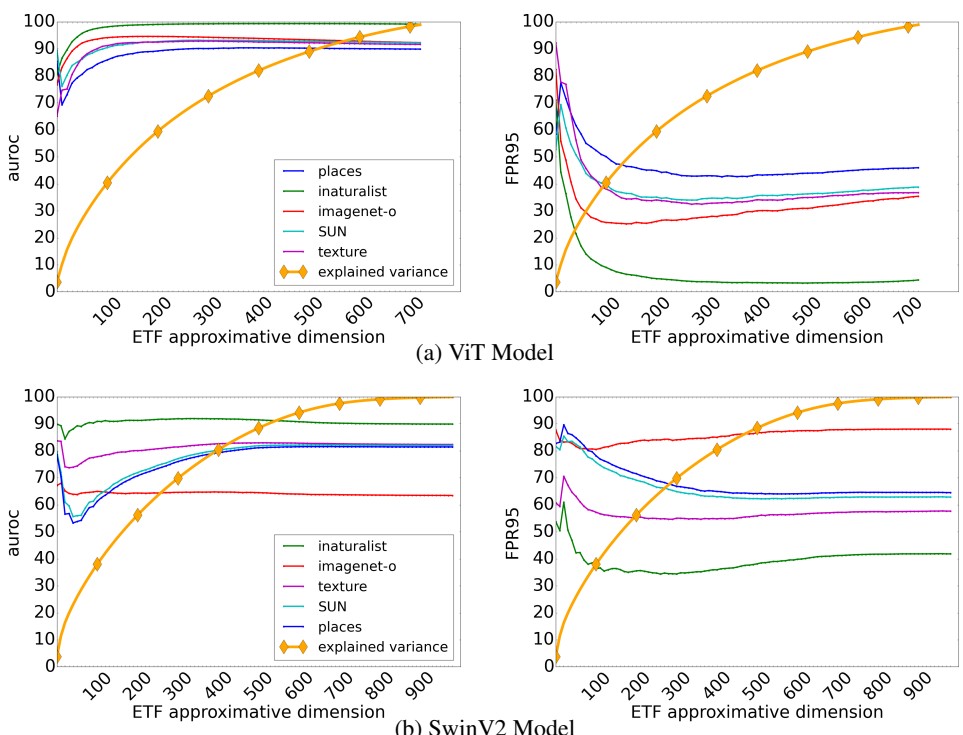

Figure C.4: Comparison of Performance metrics — AUROC (left) and FPR95 (right) — against the principal space dimension, for ViT (Top) and SwinV2 (Bottom), with ImageNet as ID data and different OOD datasets: iNaturalist, ImageNet-O, Textures, SUN, Places365.

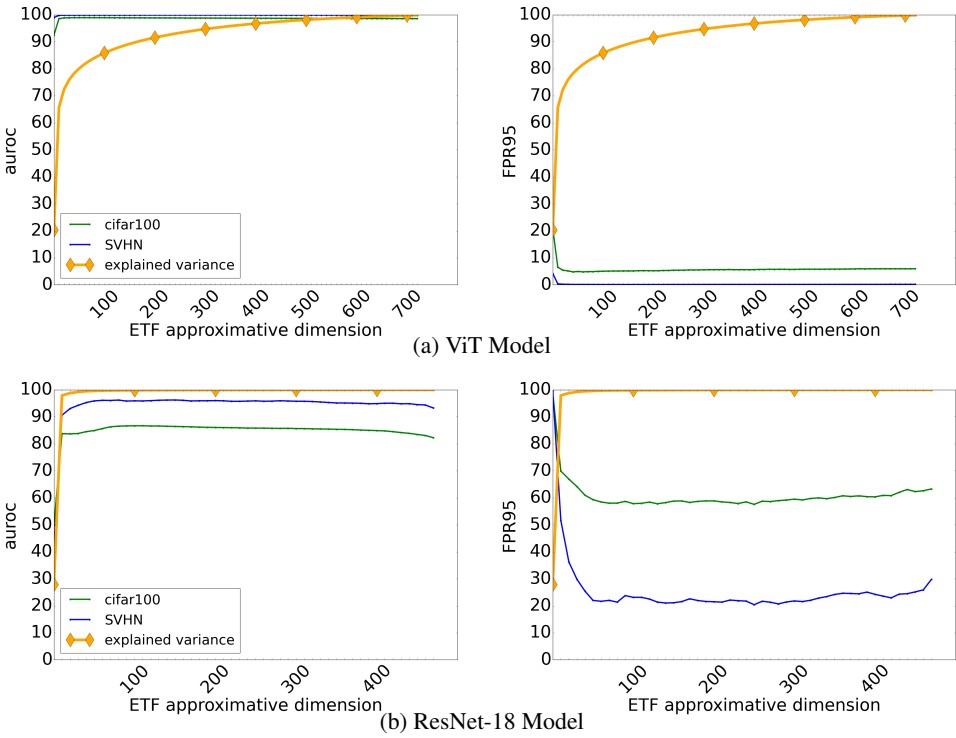

Figure C.5: Comparison of Performance metrics — AUROC (left) and FPR95 (right) — against the principal space dimension, for ViT (Top) and ResNet-18 (Bottom), with CIFAR-10 as ID data, and different OOD datasets: SVHN, CIFAR-100.

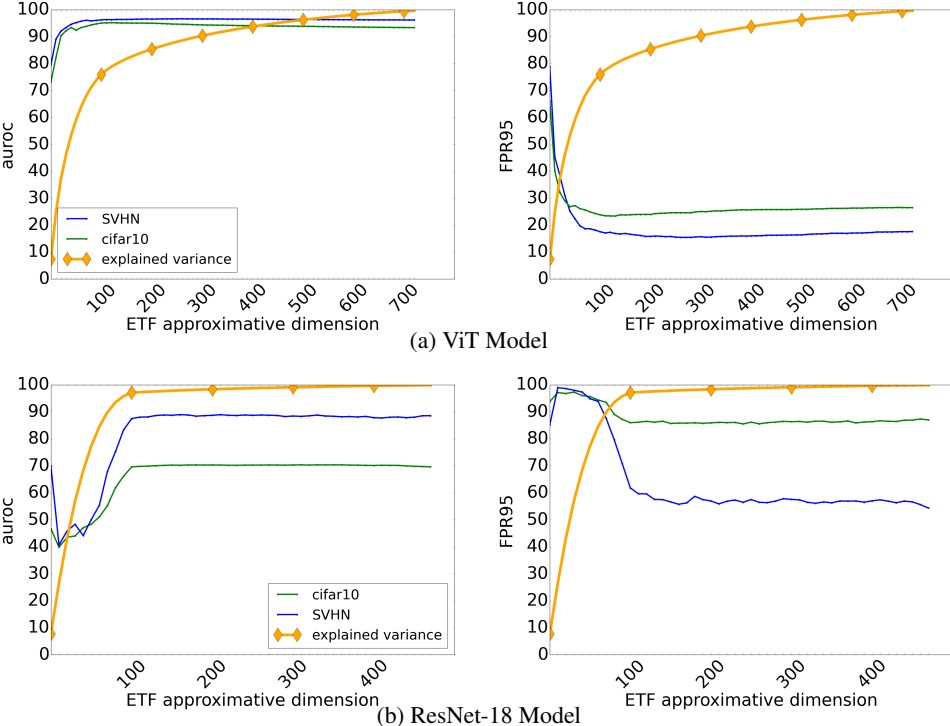

Figure C.6: Comparison of Performance metrics — AUROC (left) and FPR95 (right) — against the principal space dimension, for ViT (Top) and ResNet-18 (Bottom), with CIFAR-100 as ID data, and different OOD datasets: SVHN, CIFAR-10.

### C.6 CLASSIFICATION ACCURACY

Considering the post hoc nature of our method, when an input image is identified as in-distribution (ID), it is important to note that one can always use the original DNN classifier. This switch comes with a negligible overhead and ensures optimal performance for both classification and out-of-distribution (OOD) detection. Consequently, the classification performance of the final model remains unaltered in our approach. We refer the reader to Table C.6 for the observed classification accuracy.

| ID Data | Model | Architecture | Pre-trained dataset | Accuracy (top1%) |
|---|---|---|---|---|
| ImageNet | ViT-B/16 | Transformer | ImageNet-21k | 83.84 |
| | SwinV2-B/16 | Transformer | ImageNet-21k | 81.20 |
| | DeiT-B/16 | Transformer | ImageNet-21k | 81.80 |
| CIFAR-10 | ViT-B/16 | Transformer | ImageNet-21k | 98.89 |
| | ResNet-18 | Residual Convnet | - | 95.46 |
| CIFAR-100 | ViT-B/16 | Transformer | ImageNet-21k | 92.16 |
| | ResNet-18 | Residual Convnet | - | 78.59 |

Table C.6: Top 1% accuracy on ID data for the original classification task, for the models and ID dataset considered in our experiments, including relevant model details (from the type of architecture to with/without pretraining distinction).

## D DETAILS ABOUT NEURAL COLLAPSE

### D.1 ADDITIONAL NEURAL COLLAPSE PROPERTIES

Here, we present the equations for the remaining two properties of "neural collapse". Additionally, Figure D.7 shows a visual illustration of the Simplex ETF structure in the case of four classes. We evaluate **the convergence to Self-duality (NC3)** using the following expression:

$$\text{Self-duality} = \|\tilde{W}^\top - \tilde{M}\|_2^2 \,, \tag{7}$$

In this equation, $W$ represents the matrix of the penultimate layer classifier weights, $M$ is the matrix

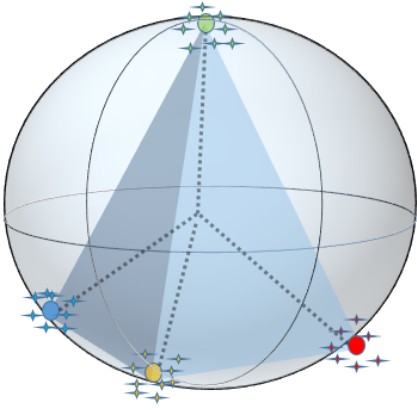

Figure D.7: Visualisation of the Simplex ETF structure.

of class means, and $[\tilde{\cdot}]$ denotes a matrix with unit-normalized means over columns. Additionally, $\|\cdot\|_2^2$ signifies the square of the $L2$ norm.

Moreover, the Simplification to Nearest Class-Center (NC4, see Equation 4) is measured as the proportion of training set samples that are misclassified when we apply the simple decision rule of assigning each feature space activation to the nearest class mean:

$$\text{Simplification-to-NCC} = \operatorname{argmin}_c \|z - \mu_c\|_2 \,, \tag{8}$$

where $z$ represents the feature vector for a given input $\boldsymbol{x}$.

### D.2 Neural collapse additional results

In this Section, we present additional results related to the convergence of the NC equations discussed in the main paper. The experiments follow the same training procedure as detailed in Section 4 and intend to give additional empirical results and theories on the neural collapse properties in the presence of OOD data.

**Convergence of NC1.** Figure D.8 presents the convergence of the NC1 variability collapse on ViT or ResNet-18 on CIFAR-10 or CIFAR-100. We observe that variability collapse on ID data tends to converge to zero, hence confirming that the DNN reaches the variability collapse in accordance with (Papyan V, 2020). In addition, we evaluate NC1 directly only on OOD data for CIFAR-100 or SVHN using a model trained on ID data. We observe that values tend to diverge or stabilize to a higher NC1 value. According to equation 1, a high value for NC1 suggests a low inter-class covariance with a high intra-class covariance. The low inter-covariance of the OOD data clusters may be interpreted as evidence that they are getting grouped together as a single cluster with a mean $\mu_G^{OOD}$, hence supporting our initial hypothesis.

Besides this, we observe that for the CIFAR-100 as ID data and CIFAR-10 as OOD data pair, the NC1 values are always relatively lower than the remainder of OOD cases. This is probably due to the fact that the CIFAR-10 label space is included entirely within the CIFAR-100 label space, hence confusing the model due to its similarity.

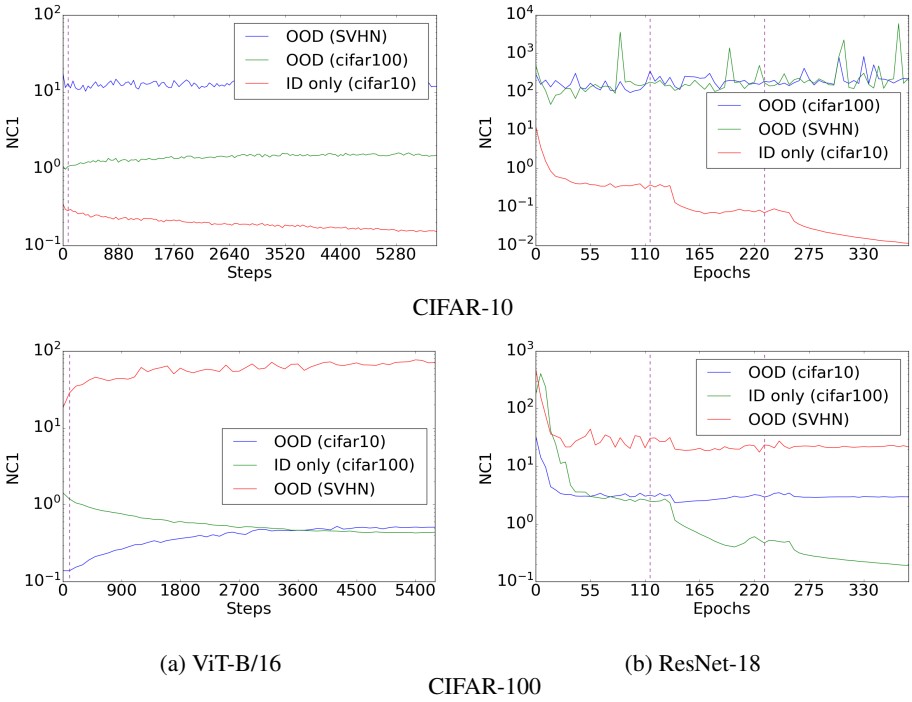

(a) ViT-B/16      (b) ResNet-18

CIFAR-100

Figure D.8: Convergence towards variability collapse (NC1) for ID/OOD data separately for ViT-B (left), ResNet-18 (right) both trained on CIFAR-100 as ID. Dashed purple lines indicate the end of warm-up steps in the case of ViT and learning rate decay epochs for ResNet-18.

**Convergence of NC2 in presence of OOD.** In order to assess "neural collapse" properties in the presence of OOD data, we investigate the convergence of NC2 equiangularity when the validation set contains OOD data. To this end, for the computation of NC2, we consider the OOD as one

supplementary class. Figure D.9 compares the convergence of NC2 when only ID data are used (in dashed black), or when OOD are included as extra classes (in blue and green). In all scenarios the NC2 value tends to converge to a plateau below 0.10. The examples with OOD data follow the same trend as those without OOD, except for experiments with CIFAR-10 as ID data that have higher NC2 values. According to NC2, this suggests that the equiangularity property between ID clusters and OOD clusters is respected, thus favoring a good separation between ID and OOD data in terms of NECO scores. We are to reinforce this observation in the next paragraph discussing NC5 convergence.

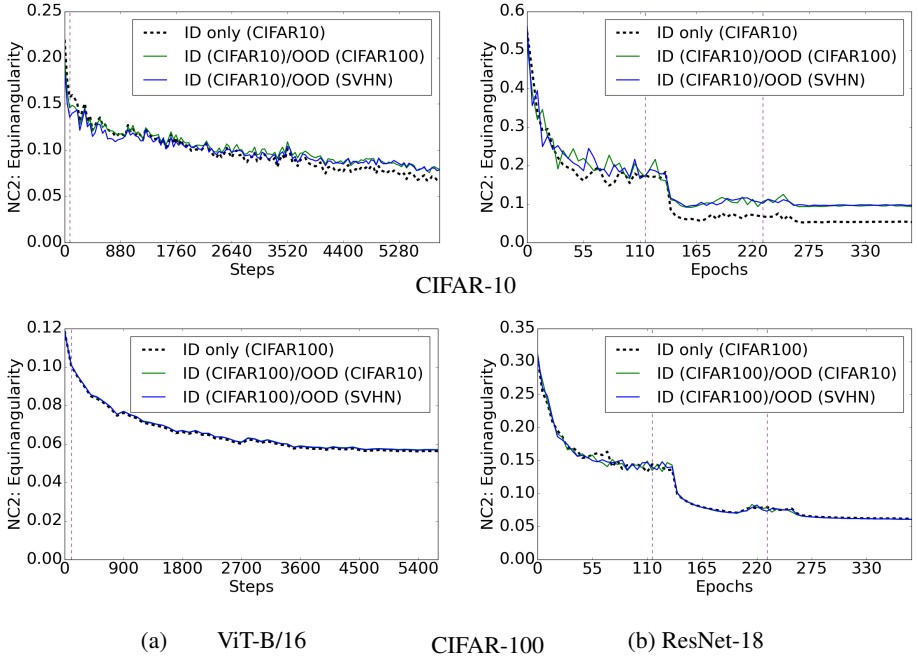

Figure D.9: Convergence of NC2 for the ID/OOD Equiangularity — in the presence of OOD data— for ViT-B (left), ResNet-18 (right) both trained on CIFAR-10 as ID. Dashed purple lines indicate the end of warm-up steps in the case of ViT and learning rate decay epochs for ResNet-18.

As for the second part of NC2, the equinormality property, its convergence on the mix of ID/OOD data is not relevant to OOD detection. Since if NC5 is verified, the norm of the OOD data will be reduced to zero in the NECO score. However, in order to guarantee an unbiased score towards any ID class, NC2 equinormality needs to be verified on the set of ID data. Since if an ID data cluster has a smaller norm than the others, it is more likely that it will be mistaken as OOD, since its closer to the null vector. Figure D.10 shows the convergence of this property on our ID data, hence guaranteeing an unbiased score.

**Convergence of NC5.** Figure D.11 illustrates the convergence of NC5 when CIFAR-10 or CIFAR-100 are used as ID datasets. In all cases, we observe for both models ( *i.e.*, ViT-B/16 and ResNet-18) very low values for NC5 (Equation 5). Hence, according to our hypothesis, the models tend to maximize the ID/OOD orthogonality as we reach the TPT. While all cases exhibit convergence of the OrthoDev, we observe that ViT tends to converge to a much lower value than ResNet-18. Hence, this may explain the performance gap between the ViT-B/16 and the ResNet-18 models.

In Section 5 we empirically demonstrate that ViT-B/16 (pre-trained on ImageNet-21K) with NECO outperfoms all baseline methods. In Figure D.12 we illustrate the NC5 OrthoDev during the fine-tuning process on ImageNet-1K against different OOD datasets (see Section 5 for the dataset details). Despite a very low initial value ($2 \times 10^{-1}$ for the worst example), we observe that NC5 tends to slightly decrease. We believe that the low initial value is due to the pretraining on the

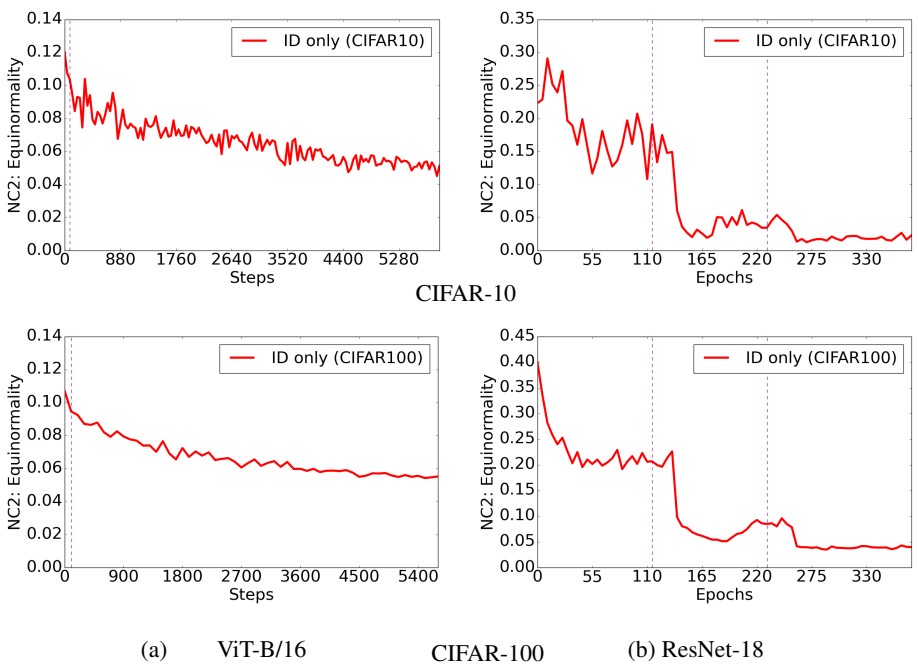

Figure D.10: Convergence of NC2 equinormality on ID data for ViT-B (left), ResNet-18 (right) both trained on CIFAR-10 as ID. Dashed purple lines indicate the end of warm-up steps in the case of ViT and learning rate decay epochs for ResNet-18.

ImageNet-21K dataset, hence favoring a premature TPT in the fine-tuning phase.

In Figure D.13, We investigate the neural collapse when the DNN is trained without pre-trained weights, and show the convergence of the NC1, NC2 and NC5. We observe that the curves tend to be steeper than experiments with pre-trained weights, reinforcing the hypothesis about the contribution of the pretraining towards reaching the TPT faster. While we are able to reach reasonable performance in the OOD cases, we observe that the NC5 value is slightly higher than experiments with pretrained weights, suggesting their contributions in the neural collapse and its properties for the OOD detection. We hypothesize that this is due to the rich information contained in the pre-trained weights that indirectly contributes to the separation in the ETF of the ID from the OOD.

**Complementary Feature Projection on Various OOD Datasets**    Finally, to further demonstrate the broad pertinence of NECO score, Figure D.14 shows features projection of ID (CIFAR-10) and OOD (CIFAR-100, iNaturalist, SUN, Places365) datasets on the first two principal components of a PCA, fitted on CIFAR-10 ID data. Each class of the ID dataset is colored and the OOD datasets are in gray. As a result of the convergence of the NC1 property, notice how the ID classes are well clustered despite the low dimensional representation. Conversely, the OOD samples are all centered at the null vector position. This suggests that the NECO score for all these cases will be highly separable between ID and OOD data, as we are to discuss in Appendix E.

# E    COMPLEMENTARY RESULTS

In this section, we present additional results that extend our findings, with either complementary DNN architectures or OOD benchmarks. The intent is to gain a better understanding of the versatility of our proposed approach, **NECO**, against different cases.

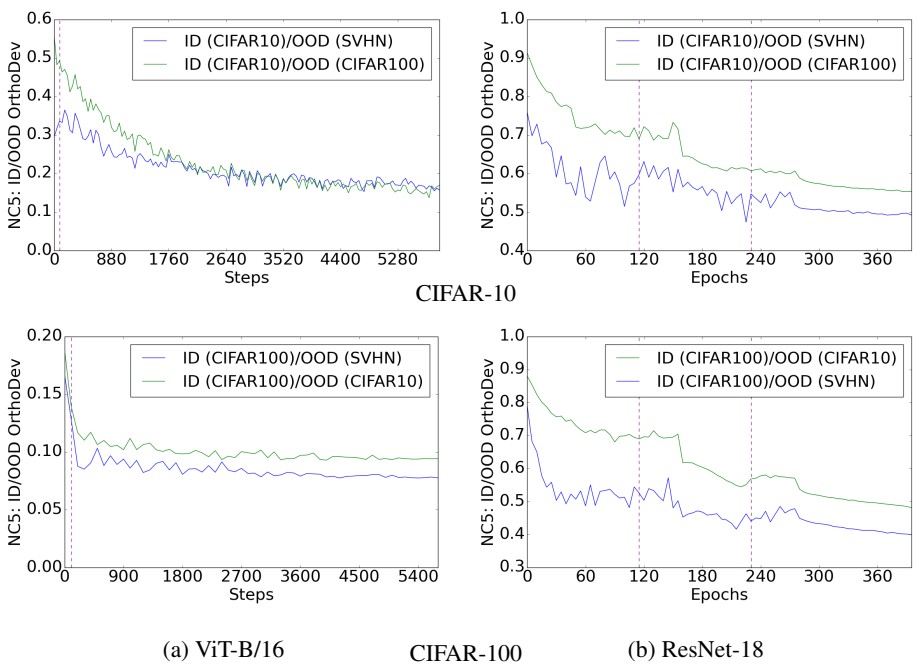

Figure D.11: Convergence of NC5 for the ID/OOD orthogonality— in the presence of OOD data — for ViT-B (left), ResNet-18 (right) both trained on CIFAR-10 (Top) and CIFAR-100 (Bottom) as ID. Dashed purple lines indicate the end of warm-up steps in the case of ViT and learning rate decay epochs for ResNet-18.

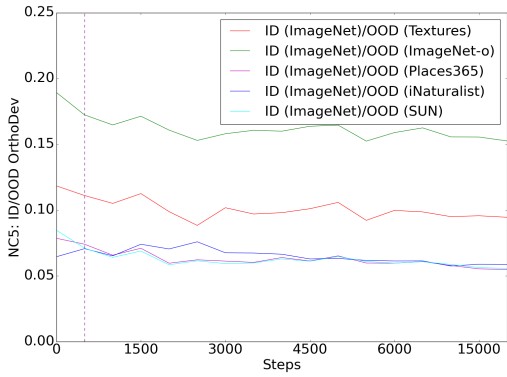

Figure D.12: Convergence towards OrthoDev minimization, for a ViT-B/16 model pre-trained on ImageNet-1K and finetuned on ImageNet-1k. Dashed purple lines indicate the end of the warm-up steps.

**NECO with DeiT and ResNet-50.** Table E.7 presents the results for NECO applied on the DeiT model fine-tuned on ImageNet-1K and evaluated on different datasets against baseline methods, as well as ResNet-50. The best performances are highlighted in bold. Contrary to the results with ViT, we observe that on average NECO with DeiT only surpasses the baselines in terms of FPR95. Whilst on ResNet-50, NECO ranks as the third best method.

**OpenOOD Benchmark — ImageNet-1K Challenge.** OpenOOD[1] is a recent test bench focusing on OOD detection proposing classic test cases from the literature. In this paragraph we focus on

---

[1]Official GitHub page of OpenOOD: https://github.com/Jingkang50/OpenOOD

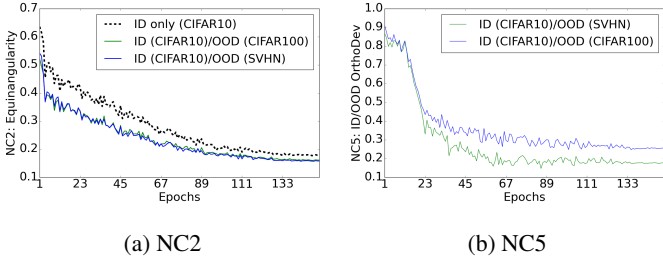

(a) NC2            (b) NC5

Figure D.13: Convergence of variability collapse (NC1), ID/OOD Equiangularity (NC2), and ID/OOD orthogonality (NC5) in the presence of OOD data. A ViT-B/16 is trained (150 epochs) without pre-training on CIFAR-10 , with Top 1% accuracy=72.07%. OOD Performance on CIFAR-10/CIFAR-100: 95.96% AUROC and 13.86% FPR95; on CIFAR-10 vs SVHN:AUROC =99.15% and FPR95 =3.77%

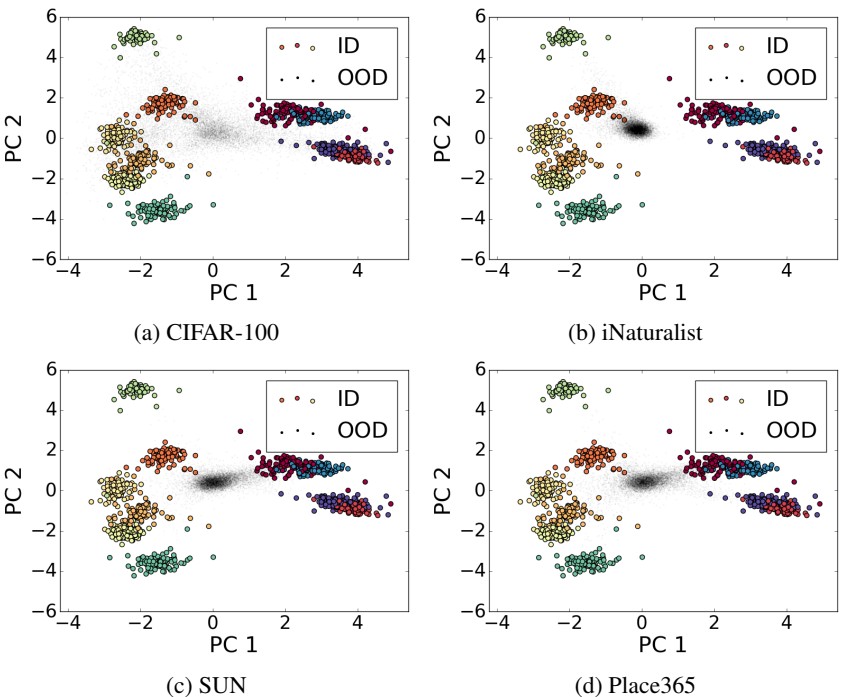

Figure D.14: Projections on the first two principal components of a PCA fitted on CIFAR-10 using the ViT penultimate layer representation. The colored points indicate the ID dataset classes while the gray points are the OOD.

| Model | Method | ImageNet-O | | Textures | | iNaturalist | | SUN | | Places365 | | Average | |
|---|---|---|---|---|---|---|---|---|---|---|---|---|---|
| | | AUROC↑ | FPR↓ | AUROC↑ | FPR↓ | AUROC↑ | FPR↓ | AUROC↑ | FPR↓ | AUROC↑ | FPR↓ | AUROC↑ | FPR↓ |
| DeiT-B/16 | Softmax score | 63.50 | 87.70 | 81.56 | 65.18 | 88.44 | 51.79 | 81.18 | 67.64 | **80.57** | 69.68 | 79.05 | 68.39 |
| | MaxLogit | 61.30 | 85.05 | 80.32 | 62.15 | 85.28 | 52.99 | 76.70 | 67.53 | 75.74 | 69.88 | 75.86 | 67.52 |
| | Energy | 60.59 | 83.80 | 77.64 | 65.27 | 78.40 | 65.52 | 71.23 | 74.99 | 69.70 | 76.53 | 71.51 | 73.22 |
| | Energy+ReAct | 64.24 | **82.30** | 80.25 | 63.94 | 84.05 | 59.69 | 77.19 | 69.54 | 75.78 | 71.41 | 76.30 | 69.37 |
| | ViM | 73.93 | 89.20 | 78.62 | 85.67 | 88.75 | 71.68 | 77.70 | 82.60 | 76.66 | 80.04 | 79.13 | 81.83 |
| | Residual | 72.80 | 92.15 | 75.37 | 89.75 | 87.08 | 77.99 | 75.67 | 86.33 | 74.99 | 83.51 | 77.18 | 85.95 |
| | GradNorm | 32.79 | 98.60 | 38.93 | 93.85 | 27.47 | 98.78 | 31.21 | 98.13 | 30.55 | 98.39 | 32.19 | 97.55 |
| | Mahalanobis | **75.37** | 90.90 | 81.04 | 82.45 | **90.32** | 67.52 | 80.95 | 81.39 | 80.21 | 78.80 | **81.57** | 80.21 |
| | KL Matching | 68.36 | 84.25 | **82.20** | 64.63 | 89.44 | 51.81 | **82.09** | 73.20 | 81.09 | 74.94 | 80.63 | 69.76 |
| | ASH-B | 46.86 | 94.05 | 60.22 | 85.50 | 46.24 | 95.00 | 35.76 | 97.45 | 35.39 | 96.91 | 44.89 | 93.78 |
| | ASH-P | 32.21 | 99.25 | 24.71 | 99.06 | 14.55 | 99.83 | 22.36 | 98.98 | 23.41 | 99.21 | 23.45 | 99.27 |
| | ASH-S | 37.90 | 95.65 | 32.85 | 97.94 | 19.77 | 99.65 | 24.98 | 97.97 | 27.13 | 97.74 | 28.53 | 97.79 |
| | NuSA | 54.65 | 94.85 | 53.96 | 95.46 | 72.47 | 85.31 | 50.12 | 96.28 | 50.74 | 95.47 | 56.39 | 93.47 |
| | **NECO (ours)** | 62.72 | 84.10 | 80.82 | **59.47** | 89.54 | **42.26** | 77.64 | **65.55** | 76.48 | **68.13** | 77.44 | **63.90** |
| ResNet-50 | Softmax score | 53.13 | 95.65 | 77.12 | 71.28 | 84.40 | 60.95 | 78.74 | **71.03** | 76.66 | **73.41** | 77.01 | 74.46 |
| | MaxLogit | 51.19 | 94.95 | 71.26 | 74.40 | 79.88 | 68.52 | 74.05 | 75.47 | 71.30 | 77.42 | 69.54 | 78.15 |
| | Energy | 48.23 | 92.80 | 49.29 | 95.64 | 50.85 | 98.14 | 50.13 | 97.93 | 48.90 | 97.76 | 49.48 | 96.45 |
| | Energy+ReAct | 36.17 | 99.05 | 40.14 | 97.67 | 30.09 | 99.98 | 28.53 | 99.79 | 27.81 | 99.80 | 32.55 | 99.26 |
| | ViM | **72.34** | **81.75** | 84.00 | 58.57 | 84.20 | 61.42 | 66.23 | 90.24 | 63.92 | 90.96 | 74.31 | 76.59 |
| | Residual | 71.90 | 82.55 | 83.31 | 61.92 | 83.13 | 65.02 | 65.54 | 91.64 | 63.54 | 92.37 | 73.48 | 78.70 |
| | GradNorm | 39.03 | 99.00 | 37.64 | 98.22 | 30.40 | 99.98 | 29.08 | 99.79 | 28.53 | 99.73 | 32.94 | 99.34 |
| | Mahalanobis | 73.35 | 85.40 | 87.54 | 53.74 | **90.73** | 49.18 | 76.60 | 83.34 | 74.84 | 84.15 | 80.61 | 71.16 |
| | KL-Matching | 67.55 | 90.45 | 86.11 | 63.93 | 89.51 | 50.04 | **81.23** | 73.89 | **79.28** | 75.98 | **80.74** | 70.86 |
| | RankFeat Song et al. (2022) | 51.15 | 97.01 | 82.19 | 74.45 | 74.45 | 90.85 | 54.69 | 95.93 | 47.92 | 97.38 | 62.08 | 91.12 |
| | ASH-B | 43.14 | 99.40 | 46.57 | 98.37 | 36.24 | 99.99 | 32.01 | 99.55 | 29.79 | 99.75 | 37.55 | 99.41 |
| | ASH-P | 52.93 | 95.10 | 50.40 | 96.38 | 57.54 | 95.64 | 58.91 | 95.30 | 59.59 | 93.66 | 55.87 | 95.22 |
| | ASH-S | 51.08 | 93.50 | 70.78 | 73.35 | 79.57 | 68.11 | 73.26 | 75.44 | 70.21 | 78.27 | 68.98 | 77.73 |
| | NuSA | 66.06 | 89.45 | 80.86 | 64.17 | 44.60 | 98.24 | 52.66 | 97.97 | 51.59 | 98.44 | 59.15 | 89.65 |
| | **NECO (ours)** | 69.80 | 86.43 | **88.09** | 51.53 | 87.94 | 62.69 | 75.56 | 77.55 | 73.07 | 78.62 | 78.89 | 71.36 |

Table E.7: OOD detection for NECO and baseline methods. The ID dataset is ImageNet-1K, and OOD datasets are Textures, ImageNet-O, iNaturalist, SUN, and Places365. Both metrics AUROC and FPR95 are in percentage. A pre-trained DeiT-B/16 and a ResNet-50 models tested. The best method is emphasized in **bold**.

the ImageNet-1K challenge with a set of OOD test cases grouped in increasing difficulties: Near-OOD — with SSB-hard (Vaze et al., 2021) and NINCO (Bitterwolf et al., 2023), Far-OOD — with iNaturalist (Horn et al., 2017), Textures (Cimpoi et al., 2014), OpenImage-O (Wang et al., 2022) , Covariate Shift — with ImageNet-C (Hendrycks & Dietterich, 2019), ImageNet-R (Hendrycks et al., 2021a), ImageNet-V2 (Recht et al., 2019). Figure E.15 shows five examples from each of these datasets.

Table E.8 presents our results on each group, including a global average performance. On average over all the test sets, we observe that NECO outperforms all the baseline methods, with all DNN models. With SwinV2, NECO reaches the first rank, in terms of FPR95, and the top3, in terms of AUROC, on all datasets. However, on the global average, ViT with NECO is the best strategy. Although NECO perform relatively well on the Covariate Shift, in particular with SwinV2, but remains limited with ViT.

**Distribution of NECO Values.** To verify the discrepancy in the distribution of NECO scores between ID and OOD values, Figure E.16 depicts the density score histograms for the ImageNet benchmark when using ViT and Swin models. These histograms reveal a significant degree of separation in certain cases, such as when iNaturalist is used as the OOD dataset with the ViT model. However, in some other cases, like when ImageNet-O is employed as the OOD dataset with the Swin model, the separation remains less pronounced. Across all the presented scenarios, ViT-B consistently outperforms other variants of vision transformers in the OOD detection task. This observation leads us to hypothesize that ViT-B exhibits more collapse than its counterparts, although further testing is needed to draw definitive conclusions.

Similarly, figure E.17 presents the density score histograms for the CIFAR-10/CIFAR-100 benchmark when using ViT and ResNet-18 models, respectively. In certain cases, such as CIFAR-10 (ID)/SVHN (OOD) with the ViT model, these histograms reveal nearly perfect separation. However, achieving separation becomes notably more challenging in the case of CIFAR-100 (ID)/CIFAR-10 (OOD). This difficulty arises because CIFAR-10 shares labels with a subset of the CIFAR-100 dataset, causing their data clusters to be much closer. This effect is particularly pronounced when using a ResNet-18 model compared to a ViT. The superior performance of the ViT model is further highlighted by its significantly lower values across all "neural collapse" metrics.

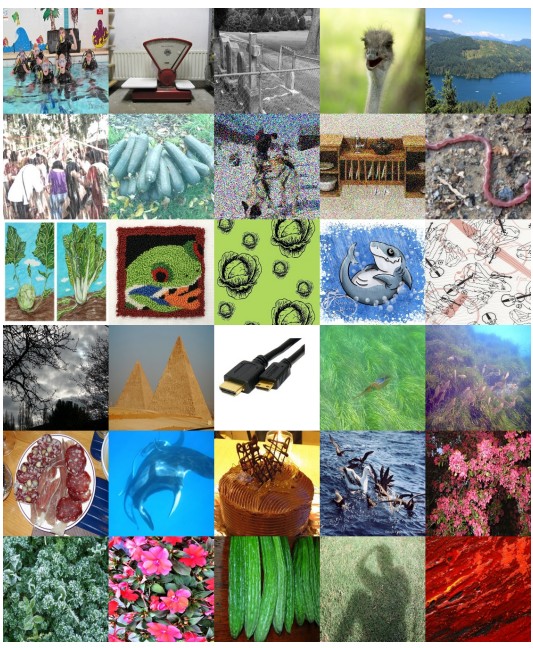

Figure E.15: Example images from OOD datasets with each row representing a dataset: ImageNet-V2,imageNet-C,imageNet-R, NINCO, SSB-hard and OpenImage-O respectively from top to bottom.

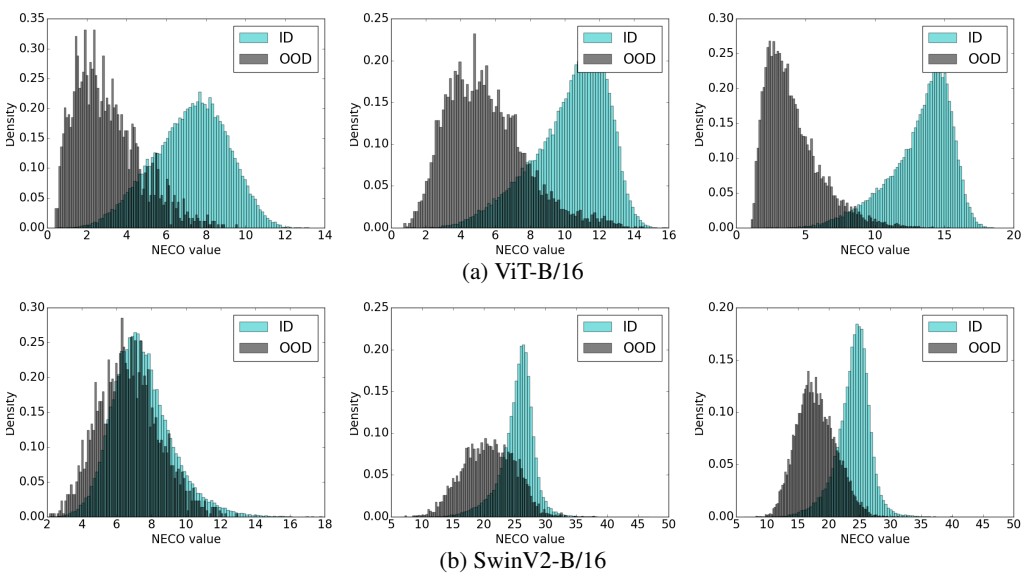

Figure E.16: ID (ImageNet) and OOD ImageNet-O (left), Textures (center) and iNaturalist (right) NECO score distributions.

| Model | Method | Covariate Shift | | | | Near OOD | | | Far OOD | | | | Global |
|---|---|---|---|---|---|---|---|---|---|---|---|---|---|
| | | ImageNet-V2 | ImageNet-C | ImageNet-R | Average | NINCO | SSB-hard | Average | Textures | iNaturalist | open-images | Average | Average |
| | | AUROC↑ FPR↓ | AUROC↑ FPR↓ | AUROC↑ FPR↓ | AUROC↑ FPR↓ | AUROC↑ FPR↓ | AUROC↑ FPR↓ | AUROC↑ FPR↓ | AUROC↑ FPR↓ | AUROC↑ FPR↓ | AUROC↑ FPR↓ | AUROC↑ FPR↓ | AUROC↑ FPR↓ |
| ViT | Softmax score | 58.61 89.37 | 71.75 71.02 | 81.42 56.82 | 70.59 72.40 | 88.97 46.67 | 80.31 66.03 | 84.64 56.35 | 86.64 49.40 | 97.21 12.14 | 93.08 29.62 | 92.31 30.39 | 82.25 52.63 |
| | MaxLogit | 58.71 88.90 | 73.64 69.46 | 86.65 47.16 | 73.00 68.51 | 92.91 35.40 | 87.28 53.10 | 90.09 44.25 | 91.69 36.90 | 98.87 5.72 | 96.75 17.33 | 95.77 19.98 | 85.81 44.25 |
| | Energy | 58.59 89.20 | 73.54 69.69 | 87.12 44.52 | 73.08 67.80 | 93.13 33.51 | 87.88 49.16 | 90.50 41.34 | 92.13 34.15 | 99.04 4.76 | 97.13 14.67 | 96.10 17.86 | 86.07 42.46 |
| | Energy+ReAct | 58.57 89.32 | 73.37 69.82 | 86.93 44.92 | 72.96 68.02 | 93.10 33.43 | 87.60 49.74 | 90.35 41.59 | 92.08 34.50 | 99.02 4.91 | 97.13 14.76 | 96.08 18.06 | 85.97 42.67 |
| | ViM | 57.33 90.38 | 70.10 80.80 | 86.26 52.61 | 71.23 74.60 | 93.12 32.36 | 88.30 45.11 | 90.71 38.73 | 91.63 38.22 | 99.59 2.00 | 97.13 15.90 | 96.12 18.71 | 85.43 44.67 |
| | Residual | 55.18 92.55 | 62.81 90.03 | 81.90 69.06 | 66.63 83.88 | 89.95 44.96 | 85.38 51.62 | 87.67 48.29 | 87.71 52.34 | 99.36 2.79 | 94.84 25.77 | 93.97 26.97 | 82.14 53.64 |
| | GradNorm | 54.45 89.64 | 69.44 68.69 | 76.54 46.91 | 66.81 68.41 | 84.21 37.14 | 80.89 51.11 | 82.55 44.12 | 86.29 35.76 | 97.45 6.17 | 93.74 16.61 | 92.49 19.51 | 80.38 44.00 |
| | Mahalanobis | 58.24 90.07 | 70.93 77.45 | 86.52 48.82 | 71.90 72.11 | 93.90 30.32 | 85.72 50.22 | 89.81 40.27 | 91.69 37.93 | 99.67 1.55 | 97.36 14.40 | 96.24 17.96 | 85.50 43.84 |
| | KL-Matching | 54.98 90.09 | 65.36 73.41 | 76.24 59.06 | 65.53 74.19 | 84.30 51.24 | 70.92 69.70 | 77.61 60.47 | 84.61 52.38 | 95.07 15.31 | 90.22 35.32 | 89.97 34.34 | 77.71 55.81 |
| | ASH-B | 48.70 94.28 | 44.13 93.79 | 57.90 89.28 | 50.24 92.45 | 57.30 89.09 | 57.23 92.61 | 57.27 90.85 | 66.05 83.29 | 72.62 80.62 | 68.34 83.42 | 69.00 82.44 | 59.03 88.30 |
| | ASH-P | 56.55 91.16 | 69.94 77.33 | 86.18 47.98 | 70.89 72.16 | 91.27 36.49 | 89.38 39.29 | 89.32 37.89 | 91.55 37.58 | 98.75 6.25 | 96.45 18.10 | 95.58 20.64 | 85.01 44.27 |
| | ASH-S | 55.88 91.29 | 68.85 78.88 | 85.23 50.25 | 69.99 73.47 | 90.34 37.67 | 89.30 37.99 | 89.82 37.83 | 91.00 39.73 | 98.50 7.28 | 95.97 19.41 | 95.16 22.14 | 84.38 45.31 |
| | NuSA | 54.63 92.51 | 63.42 90.01 | 79.82 73.19 | 65.96 85.23 | 89.97 39.66 | 87.41 45.53 | 88.69 42.59 | 88.28 51.24 | 99.30 3.12 | 94.87 25.68 | 94.15 26.68 | 82.21 52.61 |
| | **NECO (ours)** | 58.69 89.00 | 73.48 69.57 | 86.45 47.96 | 72.87 68.84 | 93.38 30.98 | 87.52 41.54 | 90.45 36.26 | 92.86 32.44 | 99.34 3.26 | 97.55 12.99 | 96.58 16.23 | 86.16 40.97 |
| SwinV2 | Softmax score | 57.76 90.14 | 75.99 67.42 | 78.35 64.50 | 70.70 74.02 | 78.54 71.91 | 70.54 82.39 | 74.54 77.15 | 81.72 60.91 | 88.59 47.66 | 85.08 57.70 | 85.13 55.42 | 77.07 67.83 |
| | MaxLogit | 56.77 90.42 | 75.41 65.83 | 76.65 63.83 | 69.61 73.36 | 74.80 73.39 | 66.68 83.64 | 70.74 78.52 | 80.36 59.55 | 86.47 50.51 | 82.45 58.75 | 83.09 56.27 | 74.95 68.24 |
| | Energy | 55.73 91.22 | 73.96 67.44 | 74.23 68.22 | 67.97 75.63 | 70.36 78.70 | 62.84 86.50 | 66.60 82.60 | 77.91 64.44 | 81.85 63.44 | 78.37 66.58 | 79.38 64.82 | 71.91 73.32 |
| | Energy+ReAct | 56.76 90.77 | 75.71 66.57 | 79.16 61.41 | 70.54 72.92 | 74.73 75.23 | 65.38 85.70 | 70.06 80.47 | 84.56 59.86 | 90.23 51.97 | 84.72 59.51 | 86.50 57.11 | 76.41 68.88 |
| | ViM | 56.29 90.88 | 68.09 82.40 | 83.09 55.67 | 69.16 76.32 | 74.55 77.97 | 65.33 87.30 | 69.94 82.64 | 81.50 61.18 | 87.54 54.09 | 85.66 55.86 | 84.90 57.04 | 75.26 70.67 |
| | Residual | 55.25 91.59 | 62.84 85.38 | 80.62 59.69 | 66.24 78.89 | 72.95 80.49 | 62.68 88.64 | 67.81 84.57 | 77.36 65.00 | 83.23 59.77 | 81.72 60.75 | 80.77 61.84 | 71.83 73.91 |
| | GradNorm | 45.78 95.52 | 48.37 86.71 | 27.00 97.20 | 40.38 93.14 | 34.90 95.34 | 40.76 95.65 | 37.83 95.50 | 33.84 93.31 | 31.82 95.01 | 29.53 95.38 | 31.73 94.57 | 36.50 94.27 |
| | Mahalanobis | 57.14 90.83 | 70.04 83.55 | 84.58 55.69 | 70.59 76.69 | 78.76 79.11 | 68.94 88.37 | 73.85 83.74 | 84.51 63.35 | 89.81 57.10 | 88.20 58.95 | 87.51 59.80 | 77.75 72.12 |
| | KL-Matching | 54.74 90.95 | 64.08 81.73 | 73.40 70.87 | 64.07 81.18 | 72.29 75.40 | 62.96 83.82 | 67.62 79.61 | 75.30 71.69 | 82.93 58.29 | 80.26 65.74 | 79.50 65.24 | 70.75 74.81 |
| | ASH-B | 50.67 95.66 | 52.07 94.49 | 50.13 97.54 | 50.96 95.90 | 53.33 96.70 | 55.91 94.86 | 54.62 95.78 | 38.59 97.50 | 48.62 97.55 | 48.00 97.75 | 45.07 97.60 | 49.66 96.51 |
| | ASH-P | 45.39 95.91 | 36.69 96.14 | 25.86 98.51 | 35.98 96.85 | 29.17 98.37 | 37.20 97.51 | 33.19 97.94 | 26.51 98.90 | 20.73 99.28 | 22.35 99.22 | 23.20 99.13 | 30.49 97.98 |
| | ASH-S | 45.17 95.35 | 36.40 94.49 | 27.52 95.78 | 36.36 95.21 | 28.09 97.23 | 33.85 97.05 | 30.97 97.14 | 36.08 94.63 | 16.15 99.50 | 22.91 97.08 | 25.05 97.07 | 30.77 96.39 |
| | NuSA | 55.99 92.21 | 53.01 93.80 | 61.16 86.98 | 56.72 90.99 | 56.70 92.19 | 54.32 93.21 | 55.51 92.70 | 62.72 83.14 | 64.01 83.58 | 64.31 82.81 | 63.68 83.17 | 59.02 88.49 |
| | **NECO (ours)** | 58.04 89.86 | 77.16 63.05 | 83.67 56.36 | 72.96 69.76 | 77.95 66.69 | 70.63 79.47 | 74.29 73.08 | 82.27 54.67 | 91.89 34.41 | 86.98 47.96 | 87.05 45.68 | 78.57 61.56 |
| DeiT | Softmax score | 58.05 90.54 | 75.42 71.27 | 78.48 64.85 | 70.65 75.55 | 79.20 72.81 | 70.56 84.10 | 74.88 78.45 | 81.56 65.18 | 88.44 51.79 | 84.64 61.14 | 84.88 59.37 | 77.04 70.21 |
| | MaxLogit | 56.84 90.33 | 75.55 67.31 | 75.40 62.66 | 69.26 73.43 | 75.08 72.18 | 67.11 84.33 | 71.09 78.25 | 80.32 62.15 | 85.28 52.99 | 80.74 60.06 | 82.11 58.40 | 70.72 69.00 |
| | Energy | 55.55 90.77 | 74.25 67.74 | 72.22 65.16 | 67.34 74.56 | 69.82 76.91 | 62.65 87.05 | 66.23 81.98 | 77.64 65.27 | 78.40 65.52 | 75.24 66.53 | 77.09 65.77 | 70.72 73.12 |
| | Energy+ReAct | 56.66 90.41 | 75.55 67.87 | 77.09 60.72 | 69.77 73.00 | 74.08 75.20 | 65.21 86.29 | 69.64 80.75 | 80.25 63.94 | 84.05 59.69 | 80.55 62.72 | 81.62 62.12 | 74.18 70.86 |
| | ViM | 56.49 90.03 | 65.68 91.97 | 84.79 59.56 | 68.99 80.52 | 78.20 81.56 | 64.45 91.12 | 71.33 86.34 | 78.62 85.67 | 88.75 71.68 | 86.40 75.08 | 84.59 77.48 | 75.42 80.83 |
| | Residual | 55.85 90.43 | 61.01 93.69 | 83.74 66.02 | 66.87 83.38 | 76.21 84.50 | 62.23 92.23 | 69.55 88.36 | 75.37 84.95 | 87.87 69.82 | 82.33 82.60 | 82.33 82.60 | 73.34 84.33 |
| | GradNorm | 46.54 94.62 | 56.39 80.16 | 26.00 98.43 | 42.98 91.07 | 34.63 97.48 | 43.45 95.91 | 39.04 96.69 | 60.22 85.50 | 46.24 95.00 | 27.40 97.89 | 44.62 92.80 | 42.61 93.12 |
| | Mahalanobis | 57.43 90.36 | 66.96 91.50 | 85.37 56.87 | 69.92 79.58 | 80.75 80.91 | 67.72 91.08 | 74.23 86.00 | 24.71 99.06 | 14.55 99.83 | 88.01 72.18 | 42.42 90.36 | 60.69 85.22 |
| | KL-Matching | 55.85 91.31 | 71.39 78.53 | 79.96 67.42 | 69.07 79.09 | 79.46 74.04 | 68.14 83.56 | 73.80 78.80 | 32.85 97.94 | 19.77 99.65 | 86.47 61.26 | 46.36 86.28 | 61.74 81.71 |
| | ASH-B | 46.88 94.53 | 40.66 97.00 | 46.99 89.40 | 44.84 93.64 | 41.14 96.50 | 34.89 98.28 | 38.02 97.39 | 38.93 93.85 | 27.47 98.78 | 40.78 94.89 | 35.73 95.84 | 39.72 95.40 |
| | ASH-P | 44.55 95.69 | 35.21 96.49 | 19.57 99.12 | 33.11 97.10 | 26.30 97.94 | 39.02 97.10 | 32.66 98.27 | 81.04 82.45 | 90.32 67.52 | 17.73 99.35 | 63.03 83.11 | 44.22 92.14 |
| | ASH-S | 48.35 94.91 | 30.08 96.88 | 41.46 95.86 | 39.96 95.88 | 31.88 97.69 | 36.75 96.91 | 34.31 97.30 | 82.20 64.63 | 89.44 51.81 | 27.41 98.04 | 66.35 71.49 | 48.45 87.09 |
| | NuSA | 52.88 93.99 | 59.16 87.85 | 59.01 93.53 | 57.02 91.79 | 64.76 86.90 | 57.30 87.31 | 61.03 87.10 | 53.96 95.46 | 72.47 85.31 | 62.41 91.49 | 62.95 90.75 | 60.12 85.29 |
| | **NECO (ours)** | 57.28 90.13 | 76.20 63.80 | 75.40 63.55 | 69.63 72.49 | 78.15 66.88 | 68.24 80.60 | 73.19 73.74 | 80.82 59.47 | 89.54 42.26 | 81.88 57.70 | 84.08 53.14 | 75.94 65.55 |

Table E.8: OOD detection for NECO against baseline methods, on the ImageNet-1K challenge from openOOD benchmark (Zhang et al., 2023) that includes: open-image, ImageNet-V2, ImageNet-R, ImageNet-C, NINCO, SSB-hard, Textures and iNaturalist. Both metrics AUROC and FPR95. are in percentage. Three architectures (ViT with fine tuning on ImageNet-1K, SwinV2, and DeiT) are evaluated. The best method is emphasized in bold, and the 2nd and 3rd ones are underlined.

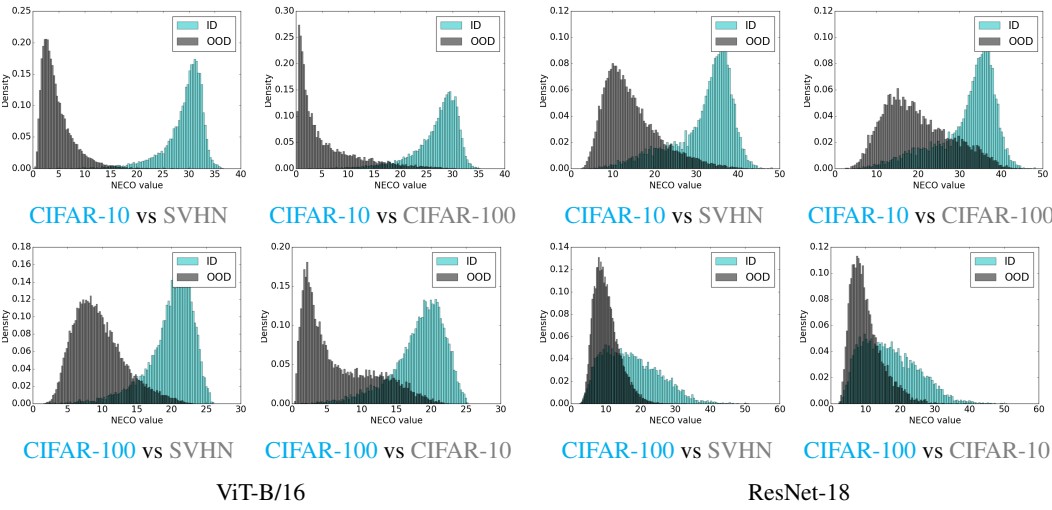

Figure E.17: NECO score distributions for different ID and OOD tuples, with a ViT-B/16 (left) and ResNet-18 (right) both fine-tuned on the ID training dataset.

