# OpenReview forum: "NECO: NEural Collapse Based Out-of-distribution detection"
_ICLR.cc/2024/Conference — ICLR 2024 poster_

### Official Review · Reviewer_Qb8y · 2023-10-27

**Soundness:** 2 fair
**Presentation:** 2 fair
**Contribution:** 2 fair
**Rating:** 6
**Confidence:** 3

**Summary:**

This work investigates the impact of Neural Collapse on the detection of OOD samples. They observe that, as training progresses,
ID and OOD data separability increases in the embedding space. Their proposed detection score computes the ratio between the norm of the projected embeddings with a PCA and the norm of the embedding. They multiply the final score with the maximum logit output to boost
performance. They run experiments with common benchmarks on image classification.

**Strengths:**

The work is well motivated with inspiration from a phenomenon studied usually exclusively for ID data: neural collapse. By revealing a novel property
in the presence of OOD data, they contributed to a better understanding of the NE phenomenon.

They also display improvements on the benchmark compared to some previous work in OOD detection.

**Weaknesses:**

**Theoretical soundness:** One of the main contributions of the paper is Theorem 4.1. But the proof is hard to follow and does not provide enough detail
(e.g., "based on the mathematical underpinnings of PCA"). Taking a more pragmatic and didactic approach could clarify the proof and strengthen the theoretical results of the paper.

**Empirical benchmark:** The benchmark on ImageNet is missing results on other popular architectures, such as ResNet, that are present only on the CIFAR
benchmark only. Results on ResNet-50 are highly appreciated and common in a plethora of previous work, and I believe would strengthen the empirical benchmark of the paper.

Some passages of the paper are hard to follow. The writing and especially the notation can be polished.

**Questions:**

1. Some notations are not clear to me. What does the operator $avg_{c,c^\prime}$ mean? $\frac{1}{c}\frac{1}{c^\prime}\sum_{i=1}^c\sum_{j=1}^{c^\prime}(\cdot)_{ij}$?
2. I would suggest the authors run experiments on ResNet-50 on the ImageNet benchmark to strengthen their empirical results.
3. What is the detection performance of only equation (8)? I suggest the authors to ablate on the influence of the MaxLogit multiplied to it.
4. In equation 6, I'm assuming the numerator goes to zero, and the denominator is always bounded. Is this correct?
5. On the proof, it states that $C\leq d \leq D$, but what if $C> D$? Which is usually the case for ImageNet-1K.
6. I may have missed it in the text, but how do the authors select the parameter $d$ for the PCA projection?

---

> ### Author Response · Authors · 2023-11-15
> **Part 1 of the answer**
>
> We appreciate your thoughtful review and constructive feedback. We have organized your concerns into seven categories and will address each of them:
>
>
>
> **Clarity in Writing:**
>  Thank you for your feedback. We employed the official notations as per the ICLR template. However, to enhance clarity, we will introduce these notations in the supplementary material. If there are other clarity issues please warn us. We have replaced the word operators by operation in section D.
>
> **Operator meaning:**  It is true that the operator $Avg_{c,c'}$ can be ambiguous.
>  In our case, this operator computes the equation  inside the $\left|...\right|$ over all possible combinations of the couple $(c , c’)$,
>   where $c,c’        \\in  \\{  1 \\ldots C\\}$, and then averages the result on these cases.
>
>
> **Additional Experiments on ResNet-50:**
> Thank you for your suggestion. We have conducted the necessary experiments using ResNet-50 and present the results in the table below.
> However, it's worth noting that the effectiveness of certain techniques may vary when applied to transformers.
>
> | **Model** | **Method** | **ImageNet-o** | **Texture** | **iNaturalist** | **SUN** | **Places365** | **Average** |
> |-----------|-------------------|------------|---------|-------------|-----|-----------|---------|
> | **ResNet-50** |**Softmax score**| 53.13 95.65 | 77.12 71.28 | 84.40 60.95 |78.74 **71.03** | 76.66 **73.41** | 77.01 74.46 |
> | | **Maxlogit** | 51.19 94.95 | 71.26 74.40 | 79.88 68.52 | 74.05 75.47 | 71.30 77.42 | 69.54 78.15 |
> | | **Energy** |48.23 92.80 | 49.29 95.64 | 50.85 98.14 | 50.13 97.93 | 48.90 97.76 |49.48 96.45 |
> | | **Energy+React** | 36.17 99.05 | 40.14 97.67 | 30.09 99.98 | 28.53 99.79 | 27.81 99.80| 32.55 99.26|
> | |**ViM** | **72.34 81.75** | 84.00 58.57 | 84.20 61.42 | 66.23 90.24 | 63.92 90.96 | 74.31 76.59|
> | |**Residual** |71.90 82.55|83.31 61.92 | 83.13 65.02 |65.54 91.64 | 63.54 92.37 | 73.48 78.70 |
> | |**GradNorm** | 39.03 99.00 | 37.64 98.22 | 30.40 99.98 | 29.08 99.79 | 28.53 99.73 | 32.94 99.34 |
> | |**Mahalanobis** | 73.35 85.40 | 87.54 53.74 | **90.73 49.18** | 76.60 83.34 | 74.84 84.15 | 80.61 71.16 |
> | |**KL-Matching** | 67.55 90.45 | 86.11 63.93 | 89.51 50.04 | **81.23** 73.89 | **79.28** 75.98 | **80.74** **70.86** |
> | |**ASH-B** | 43.14 99.40 |46.57 98.37 | 36.24 99.99 | 32.01 99.55| 29.79 99.75| 37.55 99.41 |
> | |**ASH-P** |52.93 95.10 | 50.40 96.38 | 57.54 95.64 | 58.91 95.30 | 59.59 93.66 | 55.87 95.22 |
> | |**ASH-S** |51.08 93.50 | 70.78 73.35 | 79.57 68.11 | 73.26 75.44 | 70.21 78.27 | 68.98 77.73|
> | | **NuSA** | 66.06 89.45 | 80.86 64.17| 44.60 98.24 |52.66 97.97 | 51.59 98.44 |59.15 89.65 |
> | | **NECO (ours)** | 69.80 86.43 | **88.09 51.53**| 87.94 62.69| 75.56 77.55 | 73.07 78.62 | 78.89 71.36|
>
>
> **Isolating Equation (8) Performance:**
> We conducted experiments specifically using only Equation 8 across various architectures and datasets. Here are the results obtained when the MaxLogit is not utilized:
>
> | **Model** | **Method** | **ImageNet-o** | **Texture** | **iNaturalist** | **SUN** | **Places365** | **Average** |
> |-----------|-------------------|-------------|-------------|-------------|-------------|-------------|-------------|
> | **ViT** | **NECO w/o Maxlogit** | 93.27 30.05 | 89.02 46.96 | **99.48 1.47** | 89.90 38.24 | 85.89 52.15 | 91.51 33.77 |
> |  | **NECO** | **94.53 25.20** | **92.86 32.44** | 99.34 3.26 | **93.15 33.98** | **90.38 42.66** | **94.05 27.51** |
> | **Swin** | **NECO w/o Maxlogit** | 63.17 87.95 | 75.52 69.63 | 90.72 36.37 | 81.59 **57.19** | 80.52 **60.77** | 78.31 62.38 |
> |  | **NECO** | **65.03 80.55** | **82.27 54.67** | **91.89 34.41** | **82.13** 62.26 | **81.46** 64.04 | **80.56 59.19** |
> | **DeiT** | NECO w/o Maxlogit | 56.21 95.80 | 60.02 96.68 | 71.41 84.49 | 57.29 84.10 | 55.54 82.74 | 60.10 88.76 |
> |  | **NECO** | **62.72 84.10** | **80.82 59.47** | **89.54 42.26** | **77.64 65.55** | **76.48 68.13** | **77.44 63.90** |
> | **ResNet-50** | **NECO w/o Maxlogit** | **71.01 85.44** | 84.86 60.79 | 81.04 84.85 | 60.42 96.77 | 61.92 96.34 | 71.85 84.83 |
> |  | **NECO** | 69.80 86.43 | **88.09 51.53** | **87.94 62.69** | **75.56 77.55** | **73.07 78.62** | **78.89 71.36** |
>
> **Understanding Equation 6:**
> I confirm to you that the denominator is bounded. According the NC1 property that guarantees that all ID class clusters must be far away from each other i.e maximal inter-class variance guarantees that  $||\mu_c||_2>\epsilon$.  As for the numerator, if the denominator is a finite value, the numerator converges to zero.

---

> > ### Author Response · Authors · 2023-11-15
> > **part 2 of the answer**
> >
> > **Handling Cases with less dimension than classes:**
> > The scenario where C>D poses a considerable challenge, as the vertices of the simplex ETF (i.e., the means of class clusters) cannot be perfectly orthogonal. Despite this, the robustness of NECO is maintained due to the fulfillment of NC1, NC2-equiangularity, and NC5 even in such challenging cases. These properties play a crucial role in preserving the separability of ID and OOD data, even when the perfect adherence to the simplex ETF structure definition is not feasible.
> > Additionally, NC3 asserts the geometric similarity between the DNN classifier (with dimension D=C) and the penultimate layer representation, where NECO is defined. Furthermore, NC4 indicates that the classifier, with a dimension of D=C, chooses the class with the closest class mean as the prediction. In essence, the maximum value of the classifier logits is expected to be inversely proportional to the distance to the nearest class mean, with a proportional rescaling. These insights prompted the introduction of the Maximum Logit multiplication to recalibrate NECO, addressing challenges in scenarios with a small feature layer dimension.
> >
> > **Selection of PCA Parameter d:**
> > Regarding sensitivity analysis, we have already investigated the metric's behavior with changes in PCA dimensionality, as outlined in Section C.5. Figures C.4-C.6 illustrate the robustness of NECO to the chosen dimension. Once the selected number of components sufficiently covers the entire ETF structure, adding more components has negligible impact on the AUROC. As might be expected, this dimensionality is model-dependent and requires re-evaluation with any DNN architecture changes.

---

> > > ### Comment · Reviewer_Qb8y · 2023-11-21
> > > **Follow-up**
> > >
> > > I thank the authors for the significant effort put into the rebuttal and running additional experiments.
> > >
> > > Most of my concerns were addressed, but a few issues still remains.
> > >
> > > 1. The hyper-parameter tuning procedure to obtain $d$ is not present in the main manuscript, nor is Section C.5 cited in the main text. The behavior of ResNet-18 and ViT seems quite different when d approaches D. Why is that (Figure C.5b)?
> > > 2. Even though the performance on Vision Transformers is strong, the mixed results on ResNet make it hard to claim state-of-the-art (Contribution #3). The presented method seems to be marginally better than existing ones.

---

> ### Author Response · Authors · 2023-11-22
> **Answer to the follow up question**
>
> Dear Reviewer,
>
> Thank you for responding to our comments, which have significantly contributed to the enhancement of our manuscript. Kindly find our responses to your questions below:
>
>
> 1.A   We empirically observed that the performances are stable across different values of $d$, by performing an exhaustive evaluation for all $d \in \]1, D\]$.
> In the case of the evaluation phase, we fixed $d$ by selecting the first $d$ principal components that explain at least 90% of the ID variance from the train dataset.
> In the paper, we updated Figure C4 and Figure C5  with the explained variance curve.
> 1.B Thank you for noticing the issue on Figure C5.b and Figure C6.b, we have updated the figures in the final manuscript. Now, the behavior is coherent with respect to other architectures.
>
>
> 2. Please note that we updated the claim, indicating that NECO demonstrates superior performance compared to SOTA methods. Specifically, NECO outperforms existing methods on ResNet-18 and ViT with CIFAR-10/CIFAR-100 as in-distribution datasets, and on ViT and SwinV2 with ImageNet as the in-distribution dataset.
> Furthermore, our performance on the openOOD benchmark is at state-of-the-art with ViT and SwinV2, achieving top performances in AUPR and FPR, while for DeiT, we exhibit SOTA performances in FPR. The only exception is ResNet-50 with ImageNet, where we rank second or third in performance.
>  We explain our poor performance on ResNet-50 by the fact that ViT is more overparameterized than ResNet-50, a property that favors Neural collapse. This overparameterization gap is further widened by the fact that ResNet models have a sparse latent space, i.e. many dead neurons [1] which can be tolerated for ID classification whilst impacting neural collapse. Empirically ResNets can be pruned up to 60% while preserving its classification performance. An additional reason that can be explored is the training procedure of both models.
>
>
> [1] Ding, X., Hao, T., Liu, J., Han, J., Guo, Y., & Ding, G. (2020). Lossless cnn channel pruning via gradient resetting and convolutional re-parameterization. arXiv preprint arXiv:2007.03260, 3.

---

> ### Author Response · Authors · 2023-11-23
>
> Dear Reviewer Qb8y,
>
> As the discussion period concludes today, we would like to inquire about your satisfaction with our responses. Given that we addressed all the weaknesses and questions, would you be open to reconsidering and possibly adjusting the score?
>
> Thank you for your time and consideration.

---

> > ### Comment · Reviewer_Qb8y · 2023-11-23
> >
> > I'm glad that our exchanges contributed to the enhancement of the manuscript. Since all my questions were addressed, I will raise my score.

---

### Official Review · Reviewer_Ek3F · 2023-10-29

**Soundness:** 3 good
**Presentation:** 3 good
**Contribution:** 3 good
**Rating:** 6
**Confidence:** 4

**Summary:**

In this paper, a novel post-hoc OOD detection method, NECO, is presented. NECO utilizes the geometric properties of neural collapse (NC) and principal component space to identify OOD data. This paper hypothesizes that NC not only affects the convergence of the loss during training but also impacts OOD detection.

**Strengths:**

1. This paper is clearly motivated and well-written.
2. This paper is innovative in the novel observation of ID/OOD orthogonality(NC5).
3. This paper provides thorough theoretical analysis.
4. Experimental results on the ImageNet and CIFAR benchmarks demonstrate that NECO is state-of-the-art. And NECO can be generalized to various backbones.

**Weaknesses:**

1. The NECO score is similar to the ViM score. The main difference is that the NECO score utilizes principal space features, while the ViM score utilizes null space features for OOD detection.
2. The manuscript lacks critical baseline comparisons, such as [1]. It is recommended that the authors conduct a comparison with it to provide a more comprehensive evaluation.

[1] Song, Yue, Nicu Sebe, and Wei Wang. "Rankfeat: Rank-1 feature removal for out-of-distribution detection." Advances in Neural Information Processing Systems 35 (2022): 17885-17898.

**Questions:**

Why not put NuSA in the baseline? Can NECO outperform NuSA?

---

> ### Author Response · Authors · 2023-11-15
>
> We thank you for taking your time to review our work and give us helpful comments. We have summarize you concerns into 3 categories and will answer to all of them.
>
>
>
> **NECO vs. ViM Score:**
>
> Our primary distinction from ViM lies in the utilization of the principal space, specifically the space defining the Simplex ETF structure, rather than the null space. Additionally, our scoring function is simpler, eliminating the need to compute training statistics for rescaling. It's worth noting that the null space is unaffected by the Neural Collapse (NC) phenomenon, and there's no assurance that more complex models, ones demonstrating further collapse, will yield higher values for Out-of-Distribution (OOD) samples in the null space to enhance separation. However, in our case, both experimental and theoretical justifications support the claim that using more collapsed models guarantees improved performance.
>
>
>
> **Lack of Baseline Comparisons:**
>
> Thank you for your suggestion. We will incorporate [1] into the manuscript in the coming days.
>
>
>
> **Absence of NuSA in Baseline:** In response to your recommendation and the insights from Review of Ek3F, we have now implemented NuSA using the same dimension as in ViM. The corresponding results are available in the table below, and we will incorporate these results into the paper.
>
>
> | **Model** | **Method** | **ImageNet-o** | **Texture** | **iNaturalist** | **SUN** | **Places365** | **Average** |
> |---|:---:|:---:|:---:|:---:|:---:|:---:|:---:|
> | **ViT** | 92.48 34.30 | 88.28 51.24 | 99.30 3.12 | 89.26 40.08 | 85.28 54.51 | 90.92 36.65 | 90.92 36.65 |
> | **Swin** | 56.50 91.95 | 62.72 83.14 | 64.01 83.58 | 55.97 91.28 | 54.44 92.71 | 58.73 88.53 | 58.72 88.53 |
> | **DieT** | 56.39 93.47 | 54.65 94.85 | 53.96 95.46 | 72.47 85.31 | 50.12 96.28 | 50.74 95.47 | 56.39 93.47 |
> | **ResNet-50** | 66.06 89.45 | 80.86 64.17 | 44.60 98.24 | 52.66 97.97 | 51.59 98.44 | 59.15 89.65 | 59.15 89.65 |

---

> ### Author Response · Authors · 2023-11-23
>
> Dear Reviewer Ek3F,
>
> As the discussion period concludes today, we would like to inquire about your satisfaction with our responses. Given that you have not identified any specific weaknesses, would you be open to reconsidering and possibly adjusting the score?
>
> Thank you for your time and consideration.

---

### Official Review · Reviewer_1ZaR · 2023-10-30

**Soundness:** 2 fair
**Presentation:** 2 fair
**Contribution:** 2 fair
**Rating:** 5
**Confidence:** 3

**Summary:**

This paper proposes an unsupervised anomaly detection method based on the phenomenon of neural network collapse, called NECO.The authors design an anomaly detection metric by exploiting several properties of neural networks during the training process, such as decreasing intraclass variance, equidistribution of class means, self-pairing of class means with classifiers, and orthogonality of class means with anomalous data. And experiments were conducted on several datasets and models to prove the effectiveness of NECO.

**Strengths:**

1.	A novel and simple anomaly detection metric is proposed that exploits the geometric nature of the neural network training process.
2.	Uses a large number of experiments on multiple datasets and models to demonstrate the effectiveness of NECO and compares it to other methods
3.	A clear and convincing theoretical analysis is provided, revealing the mechanisms and assumptions behind NECO.
4.	The article is well-structured with clear diagrams and well-defined notation.

**Weaknesses:**

1.	The article does not discuss the conditions and scope of the neural network collapse phenomenon, such as whether specific loss functions, activation functions, optimizers, etc.
2.	The article does not analyze the sensitivity of anomaly detection metrics.
3.	The article does not explore the interpretability of anomaly detection metrics, such as whether it can give the causes or characteristics of anomalous data.

**Questions:**

1.	In section 3.2, "Nervous Breakdown", could more visual explanations or visual diagrams of the nature of NC1 and NC2 be provided?
2.	Is the observation of Out of Distribution Neural Collapse mentioned in Section 4 based on some existing theoretical or experimental evidence? If it is based on experimental evidence, please provide more detailed experimental settings and results.
3.	Section 3 introduces many notations and definitions, but there seems to be no visual explanation or context for each definition. Could consideration be given to providing more context or explanation for each of the main definitions?
4.	Why was this particular form chosen to define NECO(x) in Eq. (8)? What is the intuition or theoretical basis behind it?
5.	Referring to the use of PCA for visualization, is it possible to describe in more detail the specific application of PCA, e.g. the number of principal components chosen?
6.	Are the limitations or possible failure scenarios of the proposed method further explored or analyzed?
7.	Throughout the article, is it possible to provide a complete flowchart or pseudo-code of the algorithm to help readers better understand the whole process of NECO?
8.	In experimental setup, are different network architectures or hyperparameters considered to validate the robustness of the approach?
9.	Have the authors considered the impact of different types of OOD data? For example, have the authors considered situations where OOD data is very similar to ID data?

---

> ### Author Response · Authors · 2023-11-15
>
> Thank you for dedicating time to review our work and providing valuable comments. We have categorized your concerns into nine points and will address each of them.
>
>
>
> **Lack of Discussion on Neural Network Collapse Conditions:**
>
> We acknowledge this concern and plan to include a discussion on the conditions that lead to Neural Collapse (NC) in deep neural networks (DNNs). We recognize the need for a comprehensive exploration of NC conditions to enhance the understanding of our approach.
>
>
>
> **Limited Analysis of Anomaly Detection Metrics:**
>
> We applied classic anomaly detection and OOD detection metrics. In response to the reviewer's suggestion, we will expand the discussion in Section 5 and include additional details in the supplementary section to describe these metrics more comprehensively. Regarding sensitivity analysis, we have already investigated the metric's behavior with changes in PCA dimensionality, as outlined in Section C.5.
>
> If you have particular experiments in mind for further evaluation, please provide details. Note that we currently have a single parameter, the PCA dimension, and we have already examined its sensitivity.
>
>
>
>
>
> **Absence of Interpretability in Anomaly Detection Metrics:**
>
> We're uncertain about the question's intent. If you're inquiring why we haven't delved into the interpretability of anomaly detection, it's important to note that it's not the primary focus of our method. Our objective is to present an OOD detection algorithm and elucidate its effectiveness. Our approach calculates the relative norm of the data within the Simplex ETF structure to derive the OOD-ness score. Therefore, a misclassification of OOD sample indicates that the data sample shares some similarity with some ID classes.
>
>
>
> **Request for Visual Explanations and pseudo code:**
>
> We are uncertain about the specific requirements, as we have already included figures 1, 2, D7, D9, D10, D11, D12, D13. Therefore, it's not entirely clear what the reviewer is requesting. If the suggestion is to provide visual explanations for the definitions in Section 3, we will incorporate them accordingly. We will also add the pseudo code in the supplementary section C.2.
>
>
>
> **Clarity on Out-of-Distribution Neural Collapse:** Our primary contribution lies in demonstrating experimental results, specifically illustrating NC1 to NC4, as shown in section 4 and section D. We have conducted a qualitative evaluation similar to [1], presenting the results in figures 1, 2, D7, D9, D10, D11, D12, D13. If additional evaluations are needed, please feel free to request them.
>
>
>
> **Rationale Behind NECO(x) Definition:**
>
> The motivation behind NECO is interesting. We observed that performing PCA on the latent space of the prelogit of pretrained DNN reveals the emergence of clusters in both the ID and OOD data when the dimension is reduced. Remarkably, the OOD cluster consistently appears centered around the null vector, as depicted in figures 2 and D13. Consequently, NECO(x) is small for OOD data. This phenomenon is consistent across different architectures. While we recognized a potential link to NC, we acknowledge that current literature in NC does not explain OOD, prompting us to propose a new NC property on OOD scenarios (NC5).

---

> ### Author Response · Authors · 2023-11-23
>
> Dear Reviewer 1ZaR,
>
> As the discussion period concludes today, we would like to inquire about your satisfaction with our responses. Given that you have not identified any specific weaknesses, would you be open to reconsidering and possibly adjusting the score?
>
> Thank you for your time and consideration.

---

### Official Review · Reviewer_r6r7 · 2023-10-31

**Soundness:** 2 fair
**Presentation:** 3 good
**Contribution:** 3 good
**Rating:** 6
**Confidence:** 4

**Summary:**

This paper proposes to study OOD detection from the perspective of Neural Collapse. In that regard, they proposed NECO, Neural Collapse-based OOD Detection. NECO involves calculating the relative norm of a sample within the subspace configuration adopted by the ID data (i.e., Simplex ETF). The authors show that NECO exhibits strong generalization capabilities across various network architectures.

**Strengths:**

This paper is overall well-written although the wording can somewhat be simplified (see weaknesses). The idea is quite simple and intuitive. The method is thoroughly evaluated and shows great performance against many benchmark datasets and OOD detection methods.

**Weaknesses:**

1. At times, the manuscript is a bit misleading. For instance, the sentence "Intuitively, these properties depicts the tendency of the network to maximally separate class features while minimizing the separation within them" is quite counter-intuitive. I'd suggest the authors to proofread the manuscript again to make it easier to digest.

2. Although NECO shows strong ID classification and OOD detection capabilities across aggregated benchmarks, it undeperformson standalone datasets such as ImageNet-O, Textures, and iNaturalist depending on the encoding architecture that is used. An explanation for this behavior could make a stronger case for the paper.

3. Since NECO is a post-hoc method, it is not obvious from the manuscript at what point in the training process does the neural collapse occurs for NECO to kick in. In other words, a model could still be well-trained without necessarily reaching the neural collapse phase. In that specific case, would NECO require training the model a bit more, or how would NECO's performance compare with say NuSA or ViM?

4. NECO makes a somewhat stringent assumption that the subspace of the ID (and OOD?) is linear by making use of PCA to learn the matrix $P$ spanned by the d eigenvectors corresponding to the largest d eigenvalues of the covariance matrix $H$, where $H$ represents the feature space of the ID data. I think a justification as to why this feature space is linear is important. A non-linear dimensionality reduction, or a manifold-based learning method, would be more appropriate given the fact that neural nets are non-linear in nature.

**Questions:**

Please refer to the weaknesses I highlighted in the above section.

---

> ### Author Response · Authors · 2023-11-15
>
> We appreciate your thorough review and the valuable feedback provided. We've categorized your concerns into four areas and addressed each of them:
>
>
>
> **Clarity Concerns:**
>
> We acknowledge the need for clarity improvements in the manuscript. A proofread is scheduled to rectify misleading formulations and correct any typos by next Friday.
>
>
>
> **Performance Variability:**
>
> The performance of out-of-distribution (OOD) detection is intricately tied to both the specific OOD dataset used and the underlying architecture, a characteristic common to all techniques.
>
> Take, for instance, the ImageNet-O dataset, which consists of adversarially filtered images specifically designed to challenge OOD detectors.
>
> The adversarial selection for this dataset was based on ResNet-50, explaining why ViT performs significantly better in this case compared to DeiT, which employs ResNet as a teacher, and Swin, which emulates ResNets in its architecture.
>
> In essence, ViT demonstrates superior performance on average, and this can be attributed to its tendency to exhibit more collapse due to its classical style of training. This is in contrast to other transformer models that modify their training procedures in diverse way: DeIT utilizes a distillation strategy while SwinV2 utilizes a shifted window mechanism.
>
> Exploring the impact of training and pre-training on the neural collapse phenomenon could provide valuable insights and serve to validate our assumptions. This requires a specific experimental setting, hence we suggest exploring this topic in future works.
>
>
>
> **Activation Point of NECO:**
>
> All models utilized in our experiments were trained until convergence, without any over-training, to ensure a fair comparison across all methods, including ViM and NuSA. While over-training has the potential to enhance neural collapse (NC), the observed improvement is marginal, as evident in figures 1, D.7-D.12. It's important to note that the models depicted in the CIFAR10 figures are not the ones used for testing; additional steps were added for figure generation to illustrate clear convergence.
>
> Enhancing NC metrics is likely to benefit out-of-distribution (OOD) detection, as it simplifies the model representation. This improvement would consequently enhance all methods leveraging NC properties, even indirectly. Self-supervised methods, such as those used in Foundation models or transformers, present promising avenues to increase the NC, as it inherently aims at clustering the data in the latent space.
>
> Regarding ViM/NuSa, while increased collapse might enhance performance, there are no guarantees, as components outside the ETF structure are not governed by the NC phenomenon. These components may or may not exhibit larger values for OOD.
>
>
>
>
>
> **Linearity Assumption:**
>
> Despite the inherent non-linearity of DNNs, the occurrence of the NC phenomenon simplifies this property, particularly at the penultimate layer/DNN-classifier level, where the ETF structure becomes linear by definition [see Introduction section: Convergence to Simplex ETF (NC2), and the figure D.7.
>
>
>
> This structure defines the feature representation and geometry. It's worth noting that alternative techniques like kernel PCA and manifold learning were tested, but no significant improvement in results was observed. If needed we can add a discussion on that.

---

> ### Author Response · Authors · 2023-11-23
>
> Dear Reviewer r6r7,
>
> As the discussion period concludes today, we would like to inquire about your satisfaction with our responses. Given that you have not identified any specific weaknesses, would you be open to reconsidering and possibly adjusting the score?
>
> Thank you for your time and consideration.

---

### Author Response · Authors · 2023-11-15
**general answer**

Dear Reviewer and Meta reviewer,

Thank you for serving as the Meta reviewer and reviewer for our submission. We thank you and the reviewers for the detailed comments and suggestions. We join a first revised version to this anwser:

-  Experiment on ResNet-50 Architecture (Section E)
-  Experiments using the NuSA score across all benchmarks
-  Ablation on the multiplication by the maximum logit (Section C.3)
-  Pseudo code for NECO method (Section C.2)

All the changes in the revised version are highlighted in orange.

Note that we will send the PDF by next Friday with the following change:
-   Revisions on the writing style

Sincerely,

Corresponding authors

---

### Author Response · Authors · 2023-11-18
**Manuscript revision**

Dear Reviewer and Meta Reviewer,

Thank you for serving as the Meta reviewer and reviewer for our submission. Attached, you will find a revised version of the paper incorporating changes to enhance the writing style. Do the reviewers have any updated perspectives or additional questions after reading the rebuttal?

Additionally, we plan to send a new version of PDF by next Wednesday, incorporating the following change:
Results on Rankfeat: Rank-1 feature removal for out-of-distribution detection.

Sincerely,
Corresponding authors

---

### Author Response · Authors · 2023-11-22
**General Answer -- Follow-up**

Dear Reviewers and Meta reviewer,


We appreciate your valuable feedback and constructive comments. In response to your input, we have implemented the following corrections:

We have added the fraction of variance explained in Figures C4 and Figure C5
We have modified section C.5
We have corrected the figure Figure C5.b and Figure C6.b
We have corrected the claims
We have added the results of  Rankfeat: Rank-1 on Table E.7

We have also performed the following corrections:
Experiment on ResNet-50 Architecture (Section E)
Experiments using the NuSA score across all benchmarks
Ablation on the multiplication by the maximum logit (Section C.3)
Pseudo code for NECO method (Section C.2)
Revisions on the writing style

We hope that these corrections address the reviewers' questions and enhance the overall quality of the paper.

---

### Meta-Review · Area_Chair_XsWk · 2023-12-08

**Metareview:**

This paper studies out-of-distribution detection from the perspective of neural collapse. The proposed method is simple yet effective, and can be supported by theoretical justification. Experimental results on the standard benchmarks are conducted and show competitive performances. The paper is well written and has a clear motivation. The investigation and method inspired by neural collapse are interesting and novel. Some reviewers are concerned about the empirical benchmark, such as the comparison with ViM. They acknowledged that their concerns were addressed after rebuttal. The AC agrees with the reviewers and recommend a accept. The authors should consider all the comments and include more benchmark results suggested by reviewers for a more comprehensive evaluation.

**Justification For Why Not Higher Score:**

The benchmard comparison in experiment is not comprehensive enough.

**Justification For Why Not Lower Score:**

This study is solid in general. The idea is interesting and proved to be effective.

---

### Decision · Program_Chairs · 2024-01-16

Accept (poster)